# IN DEFENSE OF PARAMETER SHARING FOR MODEL-COMPRESSION

**Aditya Desai**
Department of Computer Science
Rice University
Houston, TX 77054
apd10@cs.rice.edu

**Anshumali Shrivastava**
Department of Computer Science
Rice University & ThirdAI Corp.
Houston, TX 77054
anshumali@cs.rice.edu

## ABSTRACT

When considering a model architecture, there are several ways to reduce its memory footprint. Historically, popular approaches included selecting smaller architectures and creating sparse networks through pruning. More recently, randomized parameter-sharing (RPS) methods have gained traction for model compression at start of training. In this paper, we comprehensively assess the trade-off between memory and accuracy across RPS, pruning techniques, and building smaller models. Our findings demonstrate that RPS, which is both data and model-agnostic, consistently outperforms smaller models and all moderately informed pruning strategies, such as MAG, SNIP, SYNFLOW, and GRASP, across the entire compression range. This advantage becomes particularly pronounced in higher compression scenarios. Notably, even when compared to highly informed pruning techniques like Lottery Ticket Rewinding (LTR), RPS exhibits superior performance in high compression settings. This points out inherent capacity advantage that RPS enjoys over sparse models. Theoretically, we establish RPS as a superior technique in terms of memory-efficient representation when compared to pruning for linear models. This paper argues in favor of paradigm shift towards RPS based models. During our rigorous evaluation of RPS, we identified issues in the state-of-the-art RPS technique ROAST, specifically regarding stability (ROAST's sensitivity to initialization hyperparameters, often leading to divergence) and Pareto-continuity (ROAST's inability to recover the accuracy of the original model at zero compression). We provably address both of these issues. We refer to the modified RPS, which incorporates our improvements, as STABLE-RPS.

## 1 INTRODUCTION

An essential question in model design is that *given a memory budget for parameters, what is the best architecture one can create for a given task*. The solution would require us to consider the model's capacity, learning-friendliness, and generalization capability. Limited understanding of these aspects makes the problem hard. Traditionally, model architectures are designed with task-domain expertise, often augmented with neural architectural search (Pham et al., 2018). However, recently, it was shown by Lottery ticket hypothesis (Frankle & Carbin, 2018)(LTH) and follow-up works (Frankle et al., 2020b;a) that given a model architecture, the model architecture itself can be sparsified to obtain lighter models. If done intelligently, this sparsification can often maintain the accuracy to certain levels of sparsity. This has drawn interest to another exciting question *given a memory budget and an architecture, what is the best model we can derive from the architecture..* While pruning was traditionally applied as a post-processing operation (Mozer & Smolensky, 1988; LeCun et al., 1990; Hassibi & Stork, 1992; Han et al., 2015), LTH and follow-up works show that global pruning is an excellent technique to reduce the memory requirement of the model at the start of training. Since then, many works to reduce the memory footprint of the model at the beginning of training have focused on pruning as a critical technique of the algorithm (Lee et al., 2018; Tanaka et al., 2020; Zhang et al., 2022; Evci et al., 2020; Alizadeh et al., 2022). In this paper, we raise the question *"Is pruning the optimal method to reduce model parameters?"*, where optimality is measured in terms of model-memory and quality trade-off.

We suspect that pruning parameters to reduce its memory footprint is a harsh operation affecting the model capacity adversely. A simple illustration highlights this issue. Consider pruning a $n \times d$ embedding table. If we compress more than $d\times$ using pruning, we start getting degenerate zero embeddings. The same problem is present in pruning models with other components such as linear or convolutions. As we prune more and more parameters, we start dropping nodes and filters, which would affect model capacity. In the case of embedding tables, recent results from parameter sharing based compression(Chen et al., 2015; Desai et al., 2023; 2022) seems promising where 100GB sized embeddings were reduced to 10MB without loss of quality (Desai & Shrivastava, 2022). In this paper, we evaluate parameter sharing as an alternative to reduce the memory footprint of the model. In our embedding table example, randomized parameter sharing (RPS) is guaranteed to give non-degenerate embeddings no matter how small the memory is and unique embeddings with high probability. In the paper, we theoretically show that parameter sharing has a better memory capacity (thus accuracy) tradeoff in arguably a simple setting of linear models. We also rigorously evaluate memory-accuracy tradeoffs of RPS, which is model, initialization, and data agnostic, against various uninformed, moderately informed, and highly informed pruning strategies.

We find that RPS not only outperforms RAND but also consistently outperforms moderately informed pruning methods such as MAG, SNIP(Lee et al., 2018), GRASP(Zhang et al., 2022) and SYNFLOW(Tanaka et al., 2020), especially in high compression regimes (it is competitive in low-compression region to best out of all pruning methods). While RPS is inferior to the highly informed and computationally expensive pruning technique of Lottery Ticket Rewinding (LTR) (Frankle et al., 2020a) in low compression region, it surprisingly is significantly better than LTR in high compression region. This highlights RPS's inherent capacity advantage compared to pruning. Due to this capacity disadvantage, even a highly informed pruning (like LTR) is worse than a vanilla uninformed parameter-sharing method. While we restrict our attention to RPS, which is the model, initialization, and data-agnostic method, our results encourage research in informed parameter-sharing-based models to achieve even better memory-accuracy tradeoff.

In our endeavor to rigorously evaluate RPS, we use the SOTA RPS scheme of ROAST (Desai et al., 2023) and find two significant issues in ROAST (1) stability: The ROAST with its global-memory-sharing introduces a new hyper-parameter of initialization standard-deviation of the ROAST array. This standard deviation determines the different multiplicative factors used for each component. We find it essential to choose a standard deviation that keeps the multiplicative factors in check. Any extreme choice, such as a low standard deviation, will blow up multiplicative factors, causing divergence, and a high standard deviation will suppress multiplicative factors, causing slow learning. We provide a gradient scaling scheme that provably improves the stability of the training and removes its sensitivity to initialization standard deviation. (2) Pareto-continuity: ROAST cannot recover the accuracy of the entire model when compression is set to $1\times$ (no compression). This is a desirable property for any compression method, and pruning achieves this naturally. We propose a different mapping function that maintains the cache-efficient memory coalescing, obtains the optimal number of collisions, and also has the Pareto-continuity property - when compression is set to $1\times$, there is no collision; thus, the model with RPS is equivalent to the entire model. ROAST, along with improvements in gradient updates and mapping function, is collectively referred to as STABLE-RPS in the paper. We want to stress that without the mapping scheme used in STABLE-RPS, the theoretical results of the superiority of parameter sharing over pruning would not have been possible.

We make the following contributions,

- Identify and elucidate the shortcomings of the existing state-of-the-art parameter-sharing method ROAST about issues related to stability and Pareto-continuity.
- Introduce a set of enhancements that provably and demonstrably rectify the stability and Pareto-continuity issues of ROAST. These enhancements are collectively referred to as STABLE-RPS.
- We conduct a comprehensive evaluation of STABLE-RPS in comparison to building small models and pruning at initialization methods such as (1) uninformed pruning, (2) Moderately informed pruning, and (3) highly informed pruning. Our findings establish that uninformed parameter sharing is already competitive/superior to moderately informed pruning strategies at all compression factors and highly informed pruning at higher compression.
- We provide theoretical insights to explain why compressed models based on parameter sharing exhibit superior capacity compared to pruned models.

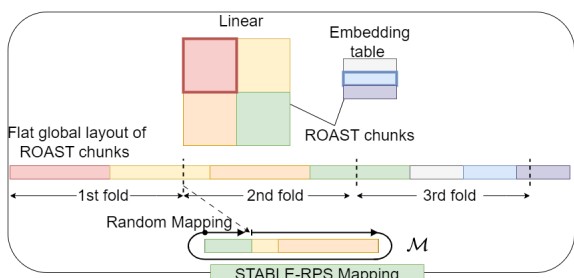

Figure 1: ROAST and STABLE-RPS mapping functions. While ROAST maps chunks independently, STABLE-RPS conceptually flattens the chunks and builds it into a single parameter array, divides it into partitions of size $m$, and independently hashes each partition.

## 2 BACKGROUND

There are various methods for model memory footprint reduction, such as designing smaller models (Iandola et al., 2016; Howard et al., 2017; Prabhu et al., 2018), low-rank component models (Jaderberg et al., 2014; Novikov et al., 2015), parameter sharing based components (Chen et al., 2015; Desai et al., 2022; 2023; 2021; Desai & Shrivastava, 2022) and sparse network (Spring & Shrivastava, 2017; Chen et al., 2020; Mocanu et al., 2018; Bellec et al., 2017). In this work, we focus on the problem of deriving a model architecture from a given full architecture, explicitly focusing on pruning and parameter sharing.

### 2.1 PARAMETER SHARING BASED METHODS

Randomized parameter-sharing(RPS) methods (Chen et al., 2015; Desai et al., 2022; 2023) disentangle the parameters from the model architecture. They maintain an array of parameters, say $\mathcal{M}$, and the weights needed inside the model are derived from this array using two hash functions, $h$ and $g$. Using hash functions $h$ and $g$ with appropriate domain, we derive the weight $w$ in the model as,

$$w = \mathcal{M}[h(id(w))] * g(id(w)) \tag{1}$$

While choice of hash functions $h$ have evolved over past few years. For $g$ we use the signed hash function. i.e., $g$ maps each weight to $\{\pm1\}$. As is clear, the size of $m = |\mathcal{M}|$ can be controlled independently, and thus, we can build models of any size. In the forward pass, the weights are derived on the fly and used for computation, whereas in the backward pass, the parameter array $\mathcal{M}$ is updated. The choice of hash functions is critical for good systemic performance. For instance, a hash function $h$ that independently maps each element will lead to many cache misses, causing system slow-down. ROAST(Desai et al., 2023) uses ROBE(Desai et al., 2022) hashing modified to the general neural network to minimize cache misses. Also, it uses a single parameter array across all model components. We discuss the working of ROAST here.

Consider a linear layer inside a model. Let the linear layer be identified by a module id $i$. ROAST divides the weight matrix inside the linear layer into chunks of size $B_K \times B_N$ where $B_K \times B_N$ are the tile sizes used by the underlying matrix multiplication kernels. Let $t(w)$ be the tile to which $w$ belongs. Any given tile $t$ inside the matrix can be identified by two integers $(x(t), y(t))$. Also, let $o(w)$ denote the row-major offset of weight $w$ in the tile $t(w)$. The hash function $h$ is defined as,

$$h(i, w) = h'(i, x(t(w)), y(t(w))) + o(w) \tag{2}$$

where $h' : \mathbf{N}^3 \rightarrow [(m - B_K B_N)]$ where $[k] = \{0, ..., k-1\}$. The mapping function essentially hashes the tile ID created with $(i, x(t(w)), y(t(w)))$, and then all the weights of the tile can be found in a contiguous chunk at this location. ROAST uses universal hash family for $h'$, i.e.,

$$h(x, y, z) = ((Ax + By + Cz + D)\%P) \tag{3}$$

where $P$ is a large prime and $A, B, C$ are randomly drawn integers from $[1, P)$ and $D$ is randomly drawn from $[0, P)$. The mapping is illustrated in the figure 1 (left).

### 2.2 PRUNING METHODS

While there are various kinds of pruning techniques - structured and unstructured, in terms of accuracy-memory tradeoff, global pruning has seen the best results (Frankle et al., 2020b). The general recipe for (global) pruning is assigning a score to each parameter and then pruning the parameters with the lowest scores. Pruning has traditionally been applied as a post-training model compression technique. However, the Lottery ticket hypothesis (LTH) used iterative magnitude pruning (IMP) to find sparse sub-networks of randomly initialized models that can achieve high

Table 1: Sensitivity of ROAST to init-stdev. (compression = $\frac{\text{original}}{\text{compressed}}$). STABLE-RPS eliminates the sensitivity to init-stdev

| Compression | | CIFAR-10(RESNET-20) | | | | | | | | | |
|---|---|---|---|---|---|---|---|---|---|---|---|
| | | Accuracy | | | | | | Norm of Solution | | | |
| | init-stdev $\rightarrow$ | 0.001 | 0.01 | 0.1 | 1 | 10 | | 0.001 | 0.01 | 0.1 | 1 | 10 |
| 1.33× | ROAST | 10.0 | 86.9 | 91.5 | 90.4 | 79.4 | | nan | 639 | 64 | 27 | 40 |
| | STABLE-RPS | 91.4 | 91.2 | 91.3 | 91.2 | 91.4 | | 56 | 56 | 56 | 56 | 56 |
| 10× | ROAST | 10.0 | 81.9 | 87.2 | 86.4 | 77.3 | | nan | 1.1k | 110 | 29 | 40 |
| | STABLE-RPS | 87.3 | 87.3 | 87.4 | 87.0 | 87.4 | | 53 | 53 | 53 | 53 | 53 |
| 100× | ROAST | 10.0 | 69.6 | 75.5 | 74.2 | 66.8 | | nan | 2.5k | 248 | 41 | 43 |
| | STABLE-RPS | 75.2 | 75.8 | 75.2 | 74.9 | 75.1 | | 48 | 47 | 48 | 48 | 48 |
| 1000× | ROAST | 10.0 | 10.0 | 53.1 | 56.0 | 52.1 | | nan | nan | 549 | 116 | 41 |
| | STABLE-RPS | 52.8 | 52.3 | 53.6 | 54.8 | 52.1 | | 45 | 45 | 45 | 46 | 45 |
| | | CIFAR-100 (VGG-11) | | | | | | | | | |
| 1.33× | ROAST | 1.0 | 67.2 | 67.5 | 63.6 | 19.8 | | 8k | 659 | 73 | 67 | 75 |
| | STABLE-RPS | 68.1 | 68.1 | 68.9 | 68.2 | 68.6 | | 115 | 115 | 115 | 115 | 109 |
| 10× | ROAST | 38.7 | 1.0 | 66.0 | 62.8 | 19.9 | | 8k | 57 | 106 | 68 | 75 |
| | STABLE-RPS | 65.8 | 66.5 | 66.1 | 66.0 | 66.3 | | 117 | 117 | 117 | 117 | 116 |
| 100× | ROAST | 1.0 | 54.5 | 57.1 | 54.4 | 20.2 | | 14k | 1741 | 192 | 70 | 75 |
| | STABLE-RPS | 57.5 | 57.4 | 57.3 | 57.4 | 58.2 | | 94 | 94 | 93 | 94 | 93 |
| 1000× | ROAST | 1.0 | 17.6 | 25.4 | 24.7 | 17.1 | | nan | 5595 | 675 | 122 | 72 |
| | STABLE-RPS | 23.5 | 23.4 | 23.7 | 23.5 | 23.6 | | 83 | 85 | 84 | 84 | 85 |

accuracy. Since then, pruning for training-cost reduction has been an active area of research. We have seen progress in LTH-based methods such as Lottery Ticket Rewinding (LTR), which, instead of re-initializing the weights to the initial weights, rewinds the weights to a later point in training (Frankle et al., 2020a). Also, a variety of single-shot pruning strategies have evolved. Most pruning strategies differ in the scoring mechanism. Some standard baselines are RAND: randomly assigns the score and MAG: assigns a score based on the absolute value of the parameter. Single-shot Network Pruning based on Connection Sensitivity (SNIP) (Lee et al., 2018) uses $|\frac{\partial \mathcal{L}}{\partial \theta} \circ \theta|$ for scoring the parameters whereas Gradient Signal Preservation(GRASP)(Zhang et al., 2022) uses $H\frac{\partial \mathcal{L}}{\partial \theta} \circ \theta$. A random sample of training data is used for this scoring. Synaptic flow pruning (SYNFLOW)(Tanaka et al., 2020) uses a scoring mechanism purely based on the weights and thus does not use any data. There have been other methods such as Alizadeh et al. (2022); Evci et al. (2020) and more. In this paper, we use MAG, SNIP, SYNFLOW and GRASP as a representative of moderately informed pruning and LTR as representative of highly informed pruning.

## 3 STABILITY OF STABLE-RPS

This section discusses the stability issues with ROAST and how to fix them. We start with empirical evidence of the sensitivity of ROAST with the hyperparameter of the standard deviation of initialization used for ROAST array ( henceforth called init-stdev). The Accuracy columns in Table 1 show that when we change the init-stdev from 1e-3 to 10, the convergence of ROAST suffers at the two extremes. When the init-stdev is very low, the optimization diverges; when the init-stdev is very high, the optimization slows down. This is consistently observed across different sparsity levels and datasets. When we look at the norms of the solutions obtained during optimization, we observe that with lower init-stdev, the norms explode for most cases, indicating divergence, and with higher init-stdev, the norms are lower, meaning slow learning. As we can see with STABLE-RPS learning scheme, which we will describe in later part of the section, we can almost eliminate the sensitivity to the init-stdev and obtain the best results at all levels of init-stdev. Thus STABLE-RPS reduces a hyper-parameter which turns out to be very sensitive towards the convergence of the RPS.

**Why is global memory sharing in RPS unstable?:** ROAST performs global memory sharing, which means that parameters from the ROAST array are shared across different components. ROAST uses multiplicative scaling factors ( say $\lambda$s) to maintain the relative scales of different components. The $\lambda$s are dependent on the init-sdev. For instance, if a component needs an initialization standard deviation of $0.1$ and the ROAST array is initialized with init-stdev of $0.01$, then the multiplicative factor $\lambda$ for this component is $0.1/0.01 = 10$. These scale factors which have to be used due to global memory sharing are the root cause of this extreme sensitivity to init-stdev.

This can be explained considering the effective change in value of parameters in a gradient descent(GD) step. Consider a parameter $x$ shared across $k$ different weights with scaling factor $\lambda_1, ..., \lambda_k$. Then, we have the recovered weights $x_i = \lambda_i x$. The effective update of $x_i$ in one

GD step can be written as follows where $x_i^{(t)}$ is recovered weight at time $t$ and $\mathcal{L}$ is the loss function.

$$(x_i)^{(t+1)} - x_i^{(t)} = \lambda_i g(i) \frac{\partial \mathcal{L}}{\partial x} = \lambda_i g(i) \sum_{j=1}^{k} \lambda_j g(j) \frac{\partial \mathcal{L}}{\partial x_j} \tag{4}$$

These $\lambda$s can have varied values. For instance, the RESNET20 model has scales varying from 1 to 5 for init-sdev of 0.0588. This implies that some parameters will see approximately $25\times$ larger effective updates than what they would see without parameter sharing. With larger effective gradients, it is unsurprising that the learning rates that worked for the full model will not work for the compressed model. Intuitively, with larger values of $\lambda$s, we will be forced to use smaller learning rates. We characterize this "stability region" of learning rates, i.e., the range of learning rate $\eta$ that can be used with guaranteed convergence of gradient descent algorithm, in the next theorem.

**Theorem 1.** *(Stability of RPS with global memory sharing) Consider a function $F$, which is $L_f$ Lipschitz smooth, with $n$ parameters. Under RPS setup with RPS array of size $m$, hashing function $h$ and scaling factors $\lambda_1, \lambda_2, ..., \lambda_n$. The stability range of gradient descent for learning rate goes from $(0, \frac{2}{L_f})$ for full model training to $\left(0, \frac{2}{\left(\max_{j=1}^{m} \sum_{i=1, h(i)=j}^{i=n} \lambda_i^2\right) L_f}\right)$ for compressed model*

The proof is presented in appendix C. Thus, the possible range of learning rates that can be used for compressed models shrinks with increasing $\lambda$s (which depends on the choice of init-stdev) and increasing compression.

**STABLE-RPS gradient scaling scheme:** We propose the following gradient scaling mechanism to remove the sensitivity to init-stdev and improve convergence. Before the update step, the gradients are scaled down using a scaler $\Gamma \in R^m$. So the update step becomes,

$$\nabla_{\text{eff}}\mathcal{L} = \Gamma \circ \nabla \mathcal{L} \tag{5}$$

where $\circ$ is element-wise multiplication. $\nabla_{\text{eff}}$ is used as the gradient in the subsequent optimizer steps. For instance in gradient descent the weights $x$ are updated as $x^{(t+1)} = x^{(t)} - \eta \nabla_{\text{eff}}\mathcal{L}$. We propose two gradient scalers. One directly follows from the above theorem 1. The other is based on maintaining effective update sizes.

We define the **theory driven scaler** as follows,

$$\Gamma[j] = \frac{1}{\sum_{i, h(i)=j} \lambda_i^2} \tag{6}$$

The gradient of a parameter $j$ from the RPS array is scaled with a factor of $l_2$ norm of all component scaling factors used for recovery from this location $j$. With this new scaling, the modified stability range of RPS with this scheme essentially remains unchanged.

**Theorem 2.** *(Stability of RPS with gradient scaling) Consider a function $F$ with $n$ parameters, which is Lipschitz smooth with constant $L_f$. Under RPS setup with RPS array of size $m$, hashing function $h$, and scaling factors $\lambda_1, \lambda_2, ...\lambda_n$. The stability range of gradient descent for learning rate is maintained at $(0, \frac{2}{L_f})$ under compression when using the gradient scaler $\Gamma[j] = \frac{1}{\sum_{i, h(i)=j} \lambda_i^2}$*

The gradient scaler effectively kills the effect of multiplicative factors. However, one can see in the proof presented in C that it is a pessimistic scaling. We find a better way to scale below.

Another way of computing scaler is to maintain effective change in value of parameters after one gradient descent step. Under the same notation of parameter $x$ being shared across $x_1, ...x_k$ using scalers $\lambda_1, ...\lambda_k$, the effective change observed in a parameter $x_i$ after the update is given in equation 4. We want to scale the change with $\gamma$ such that the effective change of all the recovered weights is similar to those with and without parameter sharing.

$$\underbrace{\frac{1}{k} \sum_{i=1}^{k} \frac{\partial \mathcal{L}}{\partial x_i}}_{\text{simple average}} \quad [\text{full}] \quad \approx \quad \frac{1}{k} \gamma \Big(\sum_{i=1}^{k} \lambda_i\Big)^2 \underbrace{\frac{\Big(\sum_{i=1}^{k} \lambda_i \frac{\partial \mathcal{L}}{\partial x_i}\Big)}{\sum_{i=1}^{k} \lambda_k}}_{\text{weighted average}} \quad [\text{RPS}] \tag{7}$$

We propose $\gamma = \frac{k}{\left(\sum_{i=1}^{k} \lambda_i\right)^2}$ and thus, the **effective update scaler** for each parameter in RPS is,

$$\Gamma[j] = \frac{\sum_{i=1, h(i)=j}^{i=n} 1}{\left(\sum_{i=1, h(i)=j}^{i=n} \lambda_i\right)^2} \forall j \in \{0, ...m-1\} \tag{8}$$

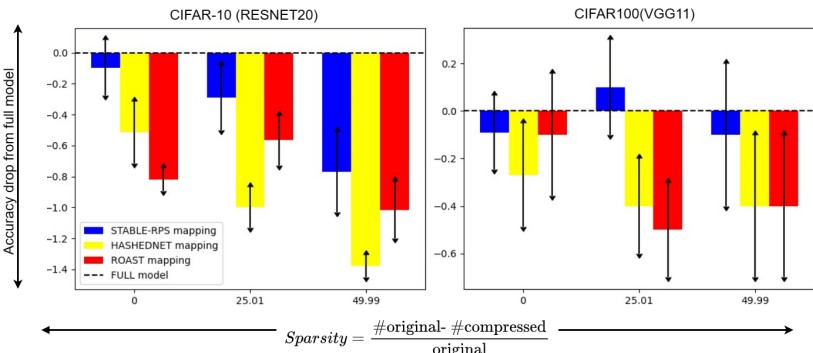

Figure 2: Pareto-continuity of STABLE-RPS, ROAST and HashedNet mapping (all mappings are used with effective update scaler described in section 3). STABLE-RPS shows Pareto-continuity and improvement over other mappings.

While both scaling mechanisms work well in stabilizing the convergence and make it robust to init-stdev, we find that an effective average-based gradient scaler works better in practice. Throughout the paper, we use this scaling method including Table 1.

## 4 PARETO-CONTINUITY OF STABLE-RPS

This section discusses the mappings used in RPS and their shortcomings concerning Pareto-continuity. Early papers on RPS used element-wise mapping where each weight is independently mapped into the weight repository. Later this idea was extended to mapping chunks to improve cache-efficiency. All these mappings lack one fundamental property that all compression techniques should have, namely *pareto-continuity* - as memory budget tends to full model memory, the mapping should revert to one-to-one mapping, causing no-collisions and thus recover exact full model (modulo permuted weight layout in memory). Pruning-based compression trivially has this property. The way randomized hashing is used in these methods, there will be some collisions even at full memory. A related issue with these mappings is that they do not have optimal *load factor*. The load factor is the maximum number of collisions under a hash function. If a hash function maps from $[n]$ to $[m]$ where $[n] = \{0, ..., n-1\}$, then optimal *load factor* is $\lceil n/m \rceil$. However, for these mappings, the *load factor* is usually much higher than optimal. A mapping that has optimal load factor will have Pareto-continuity property as well.

**STABLE-RPS mapping:** We want to build a mapping that (1) has optimal load factor ( and hence, Pareto-continuity) and (2) maintains the cache-efficiency of ROAST. An illustration of STABLE-RPS mapping is shown in the figure 1. Consider ROAST mapping. It divides the model into different-shaped chunks. We flatten out the chunks into a single array of weights. The location of each weight, say $i$, in this array is called its global index $\mathcal{G}(i)$. We perform what we call RANDOMFOLD mapping on the global index.

A RANDOMFOLD hash function $h_u : [n] \rightarrow [m]$ where $[n] = \{0, ..., n-1\}$ which uses a random hash function $u$ is defined as,

$$h_u(x) = (u(\lfloor x/m \rfloor) + (x\%m))\%m \qquad (9)$$

The RANDOMFOLD mapping divides the range $[n]$ into partitions of size $m$ and then applies a random hash function on each partition number. The mapping of a partition, then, is circular and wraps around the $m$ (see figure 1). The STABLE-RPS mapping is then defined as

$$\text{STABLE-RPS}(i) = h_u(\mathcal{G}(i)) \qquad (10)$$

We summarize the properties of STABLE-RPS in the theorem below,

**Theorem 3.** *The STABLE-RPS hash function has (1) optimal load property and, thus, Pareto-continuity (2) Under the assumption that ROAST chunks are smaller than memory budget, the number of cache-line fetches under STABLE-RPS for each ROAST chunk is $\mathcal{C}(STABLE\text{-}RPS) \leq \mathcal{C}(ROAST) + 3$ where $\mathcal{C}$ denotes number of cache-line fetches for a ROAST chunk.*

The proof is presented in appendix D. The STABLE-RPS has an optimal load factor, which leads to pareto-continuity. It pays a slight cost regarding cache efficiency due to RANDOMFOLD potentially separating ROAST chunks into different partitions.

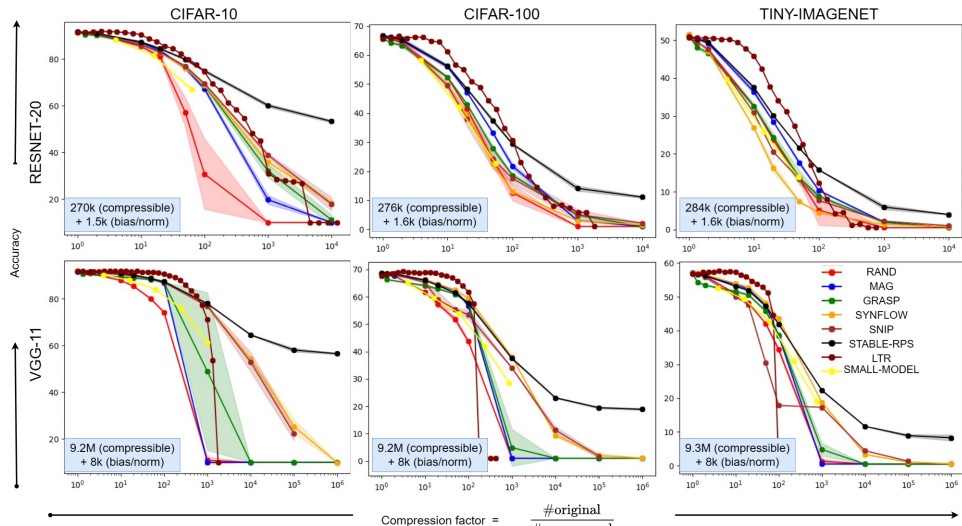

Figure 3: Memory-Accuracy tradeoff of STABLE-RPS against different pruning methods and small models for training. This figure was generated using approximately 1300 experiments. All of this tradeoff-data will be made public so that researchers can use it for plotting baselines.

In our empirical evaluation, we try the abovementioned three mappings - element-wise mapping, ROAST mapping, and STABLE-RPS mapping. We use all these mappings with gradient scaling improvements of STABLE-RPS for stable training. The results are shown in figure 2. As we can see, element-wise mapping and ROAST mapping do not achieve full model accuracy at full model memory. This is overcome by STABLE-RPS mapping. Also, STABLE-RPS mapping outperforms the other two mappings at higher compression rates as well due to its optimal load property, which minimizes the maximum collisions per bucket.

# 5 PARAMETER SHARING VS. OTHER OPTIONS

## 5.1 EMPIRICAL STUDY

We first empirically demonstrate the advantage of STABLE-RPS parameter sharing method on vision tasks. We extensively evaluate memory-accuracy tradeoffs on various model reduction techniques. The results presented in this section summarize over 1300+ experiments performed on V100 and Quadro RTX 8000 GPUs. We reestablish some known results on pruning, stress test existing results on higher and unexplored orders of compression, and add more methods to the mix. We compare the following methods.

- STABLE-RPS (uninformed): Model, data, and initialization agnostic parameter sharing method.
- RAND (uninformed): Model, data, and initialization agnostic pruning technique. In terms of information, this is equal to STABLE-RPS.
- MAG, SNIP, SYNFLOW, GRASP (Moderately informed). These techniques use data, initialization of model, or both. As pointed out by Tanaka et al. (2020), most of these methods ( except for GRASP ) benefit from iterative pruning instead of one-shot pruning. We use 100 pruning iterations for all of them except GRASP, for which we perform one-shot pruning.
- Lottery ticket Rewinding(LTR) (Highly informed) We rewind the weights to 500 iterations for all the experiments in line with the observations made by Frankle et al. (2020a). While there has been much research on finding winning tickets early, we use LTR as a representative method for the memory-accuracy tradeoff that is achievable using these methods.
- SMALL-MODEL (uninformed): To reduce the model size while using standard modules, we reduce the hidden dimensions of the model while keeping the depth same.

In Figure 3, on the x-axis we have compression factor = $\frac{\text{original}}{\text{compressed}}$. A factor of 10 means only 10% of parameters remain. We show error bars, which are $\pm$ standard deviations of accuracy. Most experimental settings (except LTR) are run on 3 or more seeds. The exact experimental details (learning rate schedules, hyper-parameters,etc.) for pruning, STABLE-RPS and LTR are provided in appendix E. We make the following observations,

**STABLE-RPS vs. Others**

- In a comparison of uninformed choices, STABLE-RPS outperforms RAND at all levels of compression. This shows that under similar information, the capacity of STABLE-RPS is strictly better than RAND pruning. We will revisit this comparison theoretically towards the end of the section.
- When compared with moderately informed pruning techniques,STABLE-RPS, which is uninformed, is always superior in higher compression regions and competitive or better in the lower-compression areas. A highly informed pruning method such as LTR is also worse in high compression regimes when compared to an uninformed STABLE-RPS. The fact that even the most informed pruning cannot beat uninformed parameter sharing at high compression highlights the severity of capacity-disadvantage pruning as a paradigm faces in comparison with STABLE-RPS.
- SMALL-MODEL performance is significantly inferior to STABLE-RPS

**Moderately informed Pruning and LTR**

- There is no one clear winner in moderately informed pruning methods. This does not contradict results presented by Tanaka et al. (2020) since we implement iterative versions of pruners, which improve over their baseline single-shot performance.
- We evaluate LTR at higher compression, which was not done before. We find that LTR experiences layer collapse, making it, at times, worse than moderately informed pruning methods. This confirms the discussion by Tanaka et al. (2020), which shows that LTR balances the weight magnitudes with expensive training, but this balancing is imperfect. It implies that LTR can be improved in high compression regions potentially using ideas from Tanaka et al. (2020).

**SMALL-MODEL vs. Others**.

- SMALL-MODEL generally gives a worse tradeoff than most moderately informed pruning and LTR. However, its tradeoff is better than RAND. This implies that a bit informed sparse networks are more powerful than constructing SMALL-MODELwith standard modules.
- The difference in the capacity of a small model and STABLE-RPS is significantly large. Also, in this case, STABLE-RPSis not data/initialization informed (unlike pruning). It shows that there is a potential for us to discover new standard modules based on parameter sharing that can be better than standard modules such as Linear and Convolutions.

## 5.2 PRUNING VS. STABLE-RPS: THEORETICAL VIEW

We now analyze the effectiveness of RAND pruning and STABLE-RPS in a theoretical setting. We first analyze the dimensionality reduction problem (preserving norms and inner products under data compression). In data compression, pruning implies dropping vector components, and STABLE-RPS means sketching vector components under the specific mapping function. We later show that the quality of learning linear models can be reduced to norm preservation under data compression. One important thing to note here is that we get these theoretical results only for STABLE-RPS mapping, and these results would not have been possible if we were analyzing other previously used mappings.

**Data compression:** Consider two vectors $x, y \in R^n$. Let the parameter budget be $m = n/k$. For the sake of simplicity, we assume that $k$ is an integer. Let $\hat{x}, \hat{y}$ be the compressed $x$ and $y$. The estimation of inner products is,

$$\langle \hat{x}, \hat{y} \rangle_{\text{prune}} = k \sum_{j=1}^{n} x_j y_j \mathbf{1}(j) \tag{11}$$

$$\langle \hat{x}, \hat{y} \rangle_{\text{STABLE-RPS}} = \sum_{i=1}^{m} \left( \sum_{j=1}^{n} x_j g(j) \mathbf{1}(h(j) = i) \right) \left( \sum_{j=1}^{n} y_j g(j) \mathbf{1}(h(j) = i) \right) \tag{12}$$

where $\mathbf{1}(i)$ is an indicator if $i^{th}$ component is sampled under pruning and $\mathbf{1}(h(j) = i)$ is an indicator if the hash mapping $h$ maps $j$ to $i$ and $g(j)$ is the sign hash function. The following theorem shows the theoretical advantage of STABLE-RPS for preserving norms. We have a detailed analysis of inner products and norms in appendix A.

**Theorem 4.** *Given vectors $x \in R^n$ under pruning and STABLE-RPS with memory budget $m = n/k$ for $k \in \mathbf{N}$, the estimates for both methods are unbiased. The variances are as follows,*

$$\mathbf{V}_{sample}(\langle \hat{x}, \hat{x} \rangle_{prune}) = \left( \frac{n}{m} - 1 \right) \sum_{i=1}^{n} x_i^4 + \left( \frac{(m-n)}{m(n-1)} \right) \sum_{i \neq j; i,j=1}^{n} x_i^2 x_j^2 \tag{13}$$

$$\mathbf{V}_h(\langle \hat{x}, \hat{x} \rangle_{STABLE-RPS}) = \frac{2}{m} \frac{n-m}{n-1} \left( \sum_{i \neq j} x_i^2 x_j^2 \right) \tag{14}$$

*Let $x_i$ and $y_i$ be i.i.d drawn from a distribution with $\mathcal{N}(\mu = 0, \sigma = \sigma_i)$. Let this data distribution be $\mathcal{D}$. Then we have,*

$$\mathbf{E}_{\mathcal{D}}(\mathbf{V}_{sample}(\langle \hat{x}, \hat{y} \rangle_{prune})) \geq \mathbf{E}_{\mathcal{D}}(\mathbf{V}_h(\langle \hat{x}, \hat{y} \rangle)_{STABLE-RPS}) \tag{15}$$

*with equality being achieved if and only if all $\sigma_i$s are equal.*

**Interpretation:** It is clear that the variance of approximation methods depends on what $x$ we are considering. So, we want to understand for what portion of data is pruning or STABLE-RPS superior. Hence, we take the expectation over data. In the case of a perfectly balanced distribution (which implies a radially symmetric distribution), the expectations are equal. So exactly the same proportions of data points favor pruning and STABLE-RPS. However, in practice, distribution is hardly symmetric. In fact, distributions are generally power-law; in this situation, STABLE-RPS is strictly better in terms of portions of data-space that it approximates better. The actual advantage will depend on the scale of power-law with a higher power-law favoring STABLE-RPS.

**Model Compression:** Now let us look at pruning and STABLE-RPS in the context of model parameters for linear models. Consider the learning of a linear model for $y$ given signal vector $x_1, ..., x_n$. Let the correlation of output variable $y$ with each $x_i$ be $\rho_i$. Let the vector $\rho = (\rho_1, \rho_2, ...\rho_n)$. We can then analyze the best residual obtained (i.e., learned model residual) once the pruning sample of weights or mapping of STABLE-RPS has been decided. The following result reduces the residual obtained under model compression to data compression on the vector $\rho$.

**Theorem 5.** *The optimal (least) residual for linear regression problem for target $y$ with standard deviation $\sigma_y$ and input signals $(x_1, x_2, ..., x_n)$ each of which has standard deviation $\sigma_x$, with correlations $\rho_i = \frac{Cov(y, x_i)}{\sigma_x \sigma_y}$, can be written as,*

$$Res_{comp}(\hat{y}) = \sigma_y^2 \left( 1 - \frac{1}{k} \langle \hat{\rho}, \hat{\rho} \rangle \right) \tag{16}$$

*where $\rho$ is the vector of all $\rho_i$s and $\langle \hat{\rho}, \hat{\rho} \rangle$ is the estimate of norm of $\rho$ vector under data compression (pruning or STABLE-RPS) and $Res_{comp}$ is residual under compression.*

**Interpretation:** The residual obtained by the compressed model (both pruning and STABLE-RPS) depends on how well the norm of the vector $\rho$ is preserved under data compression. As we have seen in the previous theorem, if the correlations are not balanced (which is usually the case in actual data), then STABLE-RPS is better than pruning on average.

## 6 DISCUSSION AND CONCLUSION

Pruning has been one of the most successful tools for model compression - post-training or at the start of training. This paper brings to light the capacity issue with sparse models obtained via pruning, which limits the success of even the most intelligent pruning schemes. On the other hand, parameter sharing is much more powerful. Uninformed parameter sharing is already better than the best pruning scheme at high compression and competitive or better than moderately informed pruning at all compression. This paper argues in favor of a paradigm shift toward parameter-sharing-based compression. Many directions need exploring to fully realize the power of parameter sharing. We briefly discuss them here. This paper talks about parameter sharing from the start of training and shows the benefits of the parameter-sharing paradigm. To widely realize the benefits of parameter sharing, we need to be able to develop a post-training parameter-sharing scheme. Another issue with parameter sharing in its current form is that it requires the same computation as the original architecture. To make it practical, we must develop implementations leveraging parameter sharing to reduce computational load. Additionally, an interesting question is whether we can leverage the power of parameter sharing to design some fundamental layer of neural network - which is low memory, low computation but high capacity. Finally, the most immediate progress in this research direction can be made if we devise informed parameter sharing strategies.

## 7 ACKNOWLEDGEMENT

This work was supported by National Science Foundation SHF-2211815, BIGDATA-1838177, ONR DURIP Grant, and grants from Adobe, Intel, Total, and VMware

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

# A   THEORY : DIMENSIONALITY REDUCTION ANALYSIS ON INNER-PRODUCTS AND NORMS

Consider two vectors $x$ and $y$ in $R^n$ space. There are two methods to perform dimensoinality reduction

1. Subsample the vector ( making it sparse / pruning ) The transformation is

$$\hat{x} = \sqrt{\frac{n}{m}} S x \tag{17}$$

   where S is a $m \times n$ sampling matrix composed of only zeros and ones. Also, each row has exactly one element non-zero. Each column has a maximum of one non-zero.

2. Projection ( randomly combining the different elements ) In this case, we perform the following transformation

$$\hat{x} = R P x \tag{18}$$

   where P is a random permutation matrix and R is a $m \times n$ matrix. R is the random STABLE-RPS matrix.

Let us look at the inner product preservation in both cases.

## A.1   PRUNING

the estimator of inner products

$$\langle \hat{x}, \hat{y} \rangle = \frac{n}{m} \sum_{i=1}^{n} x_i y_i \mathbf{1}(i) \tag{19}$$

where $\mathbf{1}(i)$ implies that $i$ is selected.

$$\mathbf{E}(\langle \hat{x}, \hat{y} \rangle) = \frac{n}{m} \frac{m}{n} \sum_{i=1}^{n} x_i y_i = \langle x, y \rangle \tag{20}$$

$$(\langle \hat{x}, \hat{y} \rangle)^2 = \frac{n^2}{m^2} \left( \sum_{i=1}^{n} x_i y_i \mathbf{1}(i) \right)^2 \tag{21}$$

$$= \frac{n^2}{m^2} \left( \sum_{i,j=1}^{n} x_i y_i x_j y_j \mathbf{1}(i) \mathbf{1}(j) \right) \tag{22}$$

$$= \frac{n^2}{m^2} \left( \sum_{i=1}^{n} x_i^2 y_i^2 \mathbf{1}(i) + \sum_{i \neq j; i,j=1}^{n} x_i y_i x_j y_j \mathbf{1}(i) \mathbf{1}(j) \right) \tag{23}$$

$$\mathbf{E}(\langle \hat{x}, \hat{y} \rangle)^2 = \left( \frac{n^2}{m^2} \frac{m}{n} \sum_{i=1}^{n} x_i^2 y_i^2 \right) + \left( \frac{n^2}{m^2} \frac{m(m-1)}{n(n-1)} \sum_{i \neq j; i,j=1}^{n} x_i y_i x_j y_j \right) \tag{24}$$

$$= \left( \frac{n}{m} \sum_{i=1}^{n} x_i^2 y_i^2 \right) + \left( \frac{n}{m} \frac{(m-1)}{(n-1)} \sum_{i \neq j; i,j=1}^{n} x_i y_i x_j y_j \right) \tag{25}$$

Thus, the variance,

$$\mathbf{V}(\langle \hat{x}, \hat{y} \rangle) = \left( \left( \frac{n}{m} - 1 \right) \sum_{i=1}^{n} x_i^2 y_i^2 \right) + \left( \left( \frac{n}{m} \frac{(m-1)}{(n-1)} - 1 \right) \sum_{i \neq j; i,j=1}^{n} x_i y_i x_j y_j \right) \tag{26}$$

$$\mathbf{V}(\langle\hat{x},\hat{y}\rangle) = \left(\left(\frac{n}{m}-1\right)\sum_{i=1}^{n}x_i^2y_i^2\right) + \left(\left(\frac{(m-n)}{m(n-1)}\right)\sum_{i\neq j;i,j=1}^{n}x_iy_ix_jy_j\right) \tag{27}$$

## A.2 PARAMETER SHARING

the estimator of inner products. Let $h$ be the final position mapping resulting from $P$ and $R$. The signs from $R$ are represented by the function $g$

$$\langle\hat{x},\hat{y}\rangle = \sum_{i=1}^{m}\left(\sum_{j=1}^{n}x_jg(j)\mathbf{1}(h(j)=i)\right)\left(\sum_{j=1}^{n}y_jg(j)\mathbf{1}(h(j)=i)\right) \tag{28}$$

where $\mathbf{1}$ is indicator. The equation can be simplified as,

$$\langle\hat{x},\hat{y}\rangle = \sum_{i=1}^{m}\left(\sum_{j,k=1}^{n}x_jy_kg(j)g(k)\mathbf{1}(h(j)=i)\mathbf{1}(h(k)=i)\right) \tag{29}$$

which can be further simplified as,

$$\langle\hat{x},\hat{y}\rangle = \sum_{i,j=1}^{n}x_iy_jg(i)g(j)\mathbf{1}(h(i)=h(j)) \tag{30}$$

$$\mathbf{E}(\langle\hat{x},\hat{y}\rangle) = \langle x,y\rangle \tag{31}$$

$$\langle\hat{x},\hat{y}\rangle^2 = \sum_{a,b,c,d}x_ay_bx_cy_dg(a)g(b)g(c)g(d)\mathbf{1}(h(a)=h(b))\mathbf{1}(h(c)=h(d)) \tag{32}$$

Only 4 cases give the non-zero terms in expectations

- $a=b=c=d$
- $a=b\neq c=d$
- $a=c\neq b=d$
- $a=d\neq b=c$

$$\mathbf{E}(\langle\hat{x},\hat{y}\rangle^2) = \sum_{i}x_i^2y_i^2 \tag{33}$$

$$+\sum_{i\neq j}x_iy_ix_jy_j \tag{34}$$

$$+\sum_{i\neq j}x_i^2y_j^2\mathbf{E}(\mathbf{1}(h(i)=h(j))) \tag{35}$$

$$+\sum_{i\neq j}x_iy_ix_jy_j\mathbf{E}(\mathbf{1}(h(i)=h(j))) \tag{36}$$

Thus variance,

$$\mathbf{V}(\langle\hat{x},\hat{y}\rangle) = \sum_{i\neq j}x_i^2y_j^2\mathbf{E}(\mathbf{1}(h(i)=h(j))) \tag{37}$$

$$+\sum_{i\neq j}x_iy_ix_jy_j\mathbf{E}(\mathbf{1}(h(i)=h(j))) \tag{38}$$

$$\mathbf{Pr}(h(i)=h(j) = \text{Pr(i and j in different chunk) Pr( i and j from different chunks collide )} \tag{39}$$

Assming that $m$ divides $n$

$$\mathbf{Pr}(h(i) = h(j)) = \left(1 - \frac{m-1}{n-1}\right)\frac{1}{m} \tag{40}$$

$$\mathbf{V}(\langle \hat{x}, \hat{y} \rangle) = \frac{1}{m}\frac{n-m}{n-1}\left(\sum_{i \neq j} x_i^2 y_j^2 + \sum_{i \neq j} x_i y_i x_j y_j\right) \tag{41}$$

### A.3 NORMS

For norms (precisely square of norm) we can just use $x = y$ in the previous analysis

$$\mathbf{V}_{prune}(\langle \hat{x}, \hat{x} \rangle) = \left(\left(\frac{n}{m} - 1\right)\sum_{i=1}^{n} x_i^4\right) + \left(\left(\frac{(m-n)}{m(n-1)}\right)\sum_{i \neq j; i,j=1}^{n} x_i^2 x_j^2\right) \tag{42}$$

$$\mathbf{V}_{STABLE-RPS}(\langle \hat{x}, \hat{x} \rangle) = \frac{2}{m}\frac{n-m}{n-1}\left(\sum_{i \neq j} x_i^2 x_j^2\right) \tag{43}$$

### A.4 COMPARISON

As is clear from the two expressions, which method of compression is better depends on the $x$ and $y$ at hand. So in order to get a sense of which method is better on "average", let us take some interesting distributions from which these points are drawn.

### A.5 AVERAGE CASE OVER DIFFERENT DISTRIBUTIONS (INNER PRODUCTS)

Let us assume that each of these $x$ and $y$ are independently drawn from distribution defined below
$$x_i \sim \mathcal{N}(0, \sigma_i^2) \tag{44}$$
In real scenarios each input may have different strengths or importance ( as we will see in the next section ) thus, we assume them to be distinct and of course, we can always get the scenario of equal strengths by setting all equal to a single $\sigma$. Let us look inner products first,

$$\mathbf{E}_D(\mathbf{V}_{prune}(\langle \hat{x}, \hat{y} \rangle)) = \left(\left(\frac{n-m}{m}\right)\sum_i \sigma_i^4\right) \tag{45}$$

$$\mathbf{E}_D(\mathbf{V}_{STABLE-RPS}(\langle \hat{x}, \hat{y} \rangle)) = \frac{1}{m}\frac{n-m}{n-1}\left(\sum_{i \neq j} \sigma_i^2 \sigma_j^2\right) \tag{46}$$

We know that,

$$\sum_{i \neq j} a_i a_j = \sum_{i<j} 2a_i a_j \leq \sum_{i<j}(a_i^2 + a_j^2) = (n-1)\sum a_i^2 \tag{47}$$

Thus,

$$\mathbf{E}_D(\mathbf{V}_{STABLE-RPS}(\langle \hat{x}, \hat{y} \rangle)) \leq \frac{1}{m}\frac{n-m}{n-1}\left((n-1)\sum_i \sigma_i^4\right) = \mathbf{E}_D(\mathbf{V}_{prune}(\langle \hat{x}, \hat{y} \rangle)) \tag{48}$$

Thus parameter sharing is strictly better than pruning and equality happens only when data is uniformly distributed. ( all standard deviations are same). Whenever there is a slight imbalance in signal strengths STABLE-RPS is superior to pruning.

A.6 AVERAGE CASE OVER DIFFERENT DISTRIBUTIONS (NORMS)

Under same distribution,

$$\mathbf{E}_{\mathcal{D}}(\mathbf{V}_{STABLE-RPS}(\langle \hat{x}, \hat{x}\rangle)) = \frac{2}{m}\frac{n-m}{n-1}\left(\sum_{i\neq j}\sigma_i^2\sigma_j^2\right) \tag{49}$$

$$\mathbf{E}_{\mathcal{D}}(\mathbf{V}_{prune}(\langle \hat{x}, \hat{x}\rangle)) = \left(\left(\frac{n}{m}-1\right)\sum_{i=1}^{n}3\sigma_i^4\right) + \left(\left(\frac{(m-n)}{m(n-1)}\right)\sum_{i\neq j;i,j=1}^{n}\sigma_i^2\sigma_j^2\right) \tag{50}$$

$$\geq \left(\left(\frac{n}{m}-1\right)3\frac{1}{(n-1)}\sum_{i\neq j}\sigma_i^2\sigma_j^2\right) + \left(\left(\frac{(m-n)}{m(n-1)}\right)\sum_{i\neq j;i,j=1}^{n}\sigma_i^2\sigma_j^2\right) \tag{51}$$

$$= \left(\left(3\frac{n-m}{m(n-1)}\right)\right) + \left(\left(\frac{(m-n)}{m(n-1)}\right)\right)\left(\sum_{i\neq j;i,j=1}^{n}\sigma_i^2\sigma_j^2\right) \tag{52}$$

$$= \left(\left(2\frac{n-m}{m(n-1)}\right)\right)\left(\sum_{i\neq j;i,j=1}^{n}\sigma_i^2\sigma_j^2\right) \tag{53}$$

$$= \mathbf{E}_{\mathcal{D}}(\mathbf{V}_{STABLE-RPS}(\langle \hat{x}, \hat{x}\rangle)) \tag{54}$$

Thus,

$$\mathbf{E}_{\mathcal{D}}(\mathbf{V}_{prune}(\langle \hat{x}, \hat{x}\rangle)) \geq \mathbf{E}_{\mathcal{D}}(\mathbf{V}_{STABLE-RPS}(\langle \hat{x}, \hat{x}\rangle)) \tag{55}$$

Thus parameter sharing is strictly better than pruning and equality happens only when data is uniformly distributed. ( all standard deviations are same). Whenever there is a slight imbalance in signal strengths STABLE-RPS is superior to pruning on average.

# B THEORY : ANALYSIS OF LINEAR MODELS UNDER PRUNING AND STABLE-RPS

Consider the following features $x_1, x_2, ...x_n$ such that,
$$Var(x_i) = V(x_i) = \sigma_x^2 \; \forall i \qquad \text{and} \qquad Cov(x_i, x_j) = C(x_i, x_j) = 0 \; \forall i \neq j \tag{56}$$

**Pruning.**
$$Res(\hat{y}) = Var(y - \sum_i \alpha_i \mathbf{1}(i)x_i) \tag{57}$$

$$Res(\hat{y}) = V(y) + \sum_i \alpha_i^2 \mathbf{1}(i)V(x_i) - 2\sum_i \alpha_i \mathbf{1}(i)C(y, x_i) \tag{58}$$

Minimizing the residual, we get,
$$\mathbf{1}(i)\alpha_i = \frac{C(y, x_i)}{\sigma_x^2} \tag{59}$$

$$Res(\hat{y}) = V(y) - \sum_i \frac{C(y, x_i)^2}{\sigma_x^2}\mathbf{1}(i) \tag{60}$$

$$Res(\hat{y}) = \sigma_y^2\left(1 - \sum_i \frac{C(y, x_i)^2}{\sigma_y^2\sigma_x^2}\mathbf{1}(i)\right) \tag{61}$$

Let $\rho(i) = \frac{Cov(y,x_i)}{\sigma_y \sigma_x}$

$$Res(\hat{y}) = \sigma_y^2 \left( 1 - \sum_i \rho(i)^2 \mathbf{1}(i) \right) \tag{62}$$

$$Res(\hat{y}) = \sigma_y^2 \left( 1 - \frac{1}{k} k \sum_i \rho(i)^2 \mathbf{1}(i) \right) \tag{63}$$

$$Res(\hat{y}) = \sigma_y^2 \left( 1 - \frac{1}{k} \langle \hat{\rho}, \hat{\rho} \rangle_{prune} \right) \tag{64}$$

where $\rho$ is the vector of all correlation coefficients where $\hat{\rho}$ is the estimation from data compression of $\rho$

**STABLE-RPS**

$$Res(\hat{y}) = Var \left( y - \sum_{j=1}^m \alpha_j \sum_i \mathbf{1}(i,j) g_i x_i \right) \tag{65}$$

$$Res(\hat{y}) = V(y) + \sum_j \left( \alpha_j^2 V \left( \sum_i \mathbf{1}(i,j) g_i x_i \right) \right) - 2 \sum_j \alpha_j C(y, \sum_i \mathbf{1}(i,j) g_i x_i) \tag{66}$$

Using the same analysis,

$$\alpha_j = \frac{C(y, \sum_i \mathbf{1}(i,j) g_i x_i)}{V(\sum_i \mathbf{1}(i,j) g_i x_i)} = \frac{\sum_i \mathbf{1}(i,j) g_i C(y, x_i)}{k \sigma_x^2} \tag{67}$$

$$Res(\hat{y}) = V(y) + \sum_j \left( \alpha_j^2 k \sigma_x^2 \right) - 2 \sum_j \alpha_j k \sigma_x^2 \alpha_j \tag{68}$$

$$Res(\hat{y}) = V(y) - \sum_j \left( \alpha_j^2 k \sigma_x^2 \right) \tag{69}$$

$$Res(\hat{y}) = \sigma_y^2 \left( 1 - \sum_j \left( \frac{\left( \sum_i \mathbf{1}(i,j) g_i C(y, x_i) \right)^2}{k \sigma_x^2 \sigma_y^2} \right) \right) \tag{70}$$

$$Res(\hat{y}) = \sigma_y^2 \left( 1 - \frac{1}{k} \sum_j \left( \left( \sum_i \mathbf{1}(i,j) g_i \frac{C(y, x_i)}{\sigma_x \sigma_y} \right)^2 \right) \right) \tag{71}$$

$$Res(\hat{y}) = \sigma_y^2 \left( 1 - \frac{1}{k} \sum_j \left( \sum_i \mathbf{1}(i,j) g_i \rho(i) \right)^2 \right) \tag{72}$$

$$Res(\hat{y}) = \sigma_y^2 \left( 1 - \frac{1}{k} \langle \hat{\rho}, \hat{\rho} \rangle_{STABLE-RPS} \right) \tag{73}$$

So the residual is minimized based on how large is the second term.

## B.1  COMPARISON

It is clear from the above two analysis that essential question is about norm-preservation of the vector of $(\rho(1), \rho(2), \rho(3), ...., \rho(n))$ under sampling and STABLE-RPS sketching. The residual in case of no compression can be obtained by just setting $\mathbf{1}(i) = 1$ for all $i$ and hence $k = 1$

$$Res(\hat{y}) = \sigma_y^2 \left(1 - \langle \rho, \rho \rangle\right) \tag{74}$$

$$\frac{\text{Res}_{comp}(\hat{y}) - \text{Res}(\hat{y})}{\text{Res}(\hat{y})} = \frac{\langle \rho, \rho \rangle - \frac{1}{k} \langle \hat{\rho}, \hat{\rho} \rangle}{1 - \langle \rho, \rho \rangle} \tag{75}$$

## B.2 Summary

We now analyse the effectiveness of RAND pruning and STABLE-RPS in theoretical setting. We first analyse the advantage in dimensionality reduction problem (preserving norms and inner products under data compression). In context of data compression, pruning implies dropping vector components and STABLE-RPS implies sketching of vector components under the specific mapping function. We later show that quality of learning of linear models can be reduced to norm preservation under data compression. One important thing to note here is that we get these theoretical results only for STABLE-RPS mapping and these results would not have been possible if we were analysing ROAST.

**Data Compression**  Consider two vectors $x, y \in R^n$. Let the parameter budget be $m = n/k$. For the sake of simplicity we assume that $k$ is an integer. Let $\hat{x}, \hat{y}$ be the compressed $x$ and $y$. The estimation of inner products is,

$$\langle \hat{x}, \hat{y} \rangle_{\text{prune}} = k \left( \sum_{j=1}^{n} x_j y_j \mathbf{1}(j) \right) \tag{76}$$

$$\langle \hat{x}, \hat{y} \rangle_{\text{STABLE-RPS}} = \sum_{i=1}^{m} \left( \sum_{j=1}^{n} x_j g(j) \mathbf{1}(h(j) = i) \right) \left( \sum_{j=1}^{n} y_j g(j) \mathbf{1}(h(j) = i) \right) \tag{77}$$

where $\mathbf{1}(i)$ is an indicator if $i^{th}$ component is sampled and $\mathbf{1}(h(j) == i)$ is an indicator if the hash mapping $h$ maps $j$ to $i$ and $g(j)$ is the sign hash function. The following theorem shows the theoretical advantage of STABLE-RPS. In order to make equations simpler, we assume that the parameters are first permuted before applying the STABLE-RPS mapping.

**Theorem 6.** *Given vectors $x, y \in R^n$ and under pruning and STABLE-RPSwith memory budget $m = n/k$ for $k \in \mathbf{N}$, the estimates for both methods are unbiased. The variances are as follows,*

$$\mathbf{V}_{sample}(\langle \hat{x}, \hat{y} \rangle_{prune}) = \left( \left( \frac{n}{m} - 1 \right) \sum_{i=1}^{n} x_i^2 y_i^2 \right) + \left( \left( \frac{(m-n)}{m(n-1)} \right) \sum_{i \neq j; i,j=1}^{n} x_i y_i x_j y_j \right) \tag{78}$$

$$\mathbf{V}_h(\langle \hat{x}, \hat{y} \rangle_{STABLE-RPS}) = \frac{1}{m} \frac{n-m}{n-1} \left( \sum_{i \neq j} x_i^2 y_j^2 + \sum_{i \neq j} x_i y_i x_j y_j \right) \tag{79}$$

*As can be seen, depending on exact vectors, either method can be better than each other. As both these methods are data agnostic, let us take a look at an average case over some data distribution. Let $x_i$ and $y_i$ be i.i.d drawn from any distribution with $\mu = 0, \sigma = \sigma_i$. Let this data distribution be $\mathcal{D}$. Then we have,*

$$\mathbf{E}_{\mathcal{D}}(\mathbf{V}_{sample}(\langle \hat{x}, \hat{y} \rangle_{prune})) \geq \mathbf{E}_{\mathcal{D}}(\mathbf{V}_h(\langle \hat{x}, \hat{y} \rangle)_{STABLE-RPS}) \tag{80}$$

*with equality being achieved if and only if all $\sigma_i$s are equal.*

We can similarly analyse preservation of norms and the results are summarized below,

**Theorem 7.** *Given vectors $x \in R^n$ and under pruning and STABLE-RPSwith memory budget $m = n/k$ for $k \in \mathbf{N}$, the estimates for both methods are unbiased. The variances are as follows,*

$$\mathbf{V}_{sample}(\langle \hat{x}, \hat{x} \rangle_{prune}) = \left( \left( \frac{n}{m} - 1 \right) \sum_{i=1}^{n} x_i^4 \right) + \left( \left( \frac{(m-n)}{m(n-1)} \right) \sum_{i \neq j; i,j=1}^{n} x_i^2 x_j^2 \right) \tag{81}$$

$$\mathbf{V}_h(\langle \hat{x}, \hat{x} \rangle_{STABLE-RPS}) = \frac{2}{m} \frac{n-m}{n-1} \left( \sum_{i \neq j} x_i^2 x_j^2 \right) \tag{82}$$

*Let $x_i$ and $y_i$ be i.i.d drawn from a distribution with $\mathcal{N}(\mu = 0, \sigma = \sigma_i)$. Let this data distribution be $\mathcal{D}$. Then we have,*

$$\mathbf{E}_{\mathcal{D}}(\mathbf{V}_{sample}(\langle \hat{x}, \hat{y} \rangle_{prune})) \geq \mathbf{E}_{\mathcal{D}}(\mathbf{V}_h(\langle \hat{x}, \hat{y} \rangle)_{STABLE-RPS}) \tag{83}$$

*with equality being achieved if and only if all $\sigma_i$s are equal.*

**Interpretation:** As pointed out in theorems, the variance really depends on what $x$ (or $x$, $y$) are we considering. So, we want to understand for what portion of points is pruning or STABLE-RPS superior. Hence, we take the expectation over data. In case of perfectly balanced distribution (which implies a radial symmetric distribution), the expectations are equal. So exactly same proportions of data points favor pruning / STABLE-RPS. However, in practice distribution is hardly symmetric. In fact distributions are generally power-law, In this situation STABLE-RPS is strictly better in terms of portions of data-space that it approximates better. The actual advantage will depend on scale of power-law with higher power-law favoring STABLE-RPS

**Model Compression** Now let us take a look at pruning and STABLE-RPSin context of model parameters for linear models. Consider the learning of a linear model for $y$ given signal vector $x_1, ..., x_n$. Let the correlation of output variable $y$ with each $x_i$ be $\rho_i$. Let the vector $\rho = (\rho_1, \rho_2, ... \rho_n)$. We can then analyse the best residual obtained (i.e. learned model residual) once the pruning sample of weights or mapping of STABLE-RPS has been decided. The following result reduces the residual to data compression on the vector $\rho$.

**Theorem 8.** *The optimal (least) residual for linear regression problem for target $y$ with standard deviation $\sigma_y$ and input signals $(x_1, x_2, ..., x_n)$ each of which has standard deviation $\sigma_x$, with correlations $\rho_i = \frac{Cov(y, x_1)}{\sigma_x \sigma_y}$, can be written as,*

$$\frac{Res_{comp}(\hat{y}) - Res(\hat{y})}{Res(\hat{y})} = \frac{\langle \rho, \rho \rangle - \frac{1}{k} \langle \hat{\rho}, \hat{\rho} \rangle}{1 - \langle \rho, \rho \rangle} \tag{84}$$

*where $\rho$ is the vector of all $\rho_i$s and $\langle \hat{\rho}, \hat{\rho} \rangle$ is the estimate of norm of $\rho$ vector under data compression (pruning or STABLE-RPS)*

**Interpretation:** As can be seen from the above theorem, the residual obtained by the compressed model (both pruning and STABLE-RPS) depends on how well the norm of the vector $\rho$ is preserved under compression. As we have seen in the previous theorem, if the correlations are not balanced (which is usually the case in real data), then STABLE-RPS is better than pruning on average.

## C  CONVERGENCE OF ROAST AND STABLE-RPS

We consider a simple case of Lipschitz continuous function with gradient descent and look at the stability region for the learning rate.

The update is

$$\theta_{t+1} = \theta_t - \eta \nabla F(\theta_t) \tag{85}$$

Using the lipschitz smoothness, we have

$$F(y) \leq F(x) + \langle \nabla F(x), y - x \rangle + \frac{L}{2} ||y - x||^2 \tag{86}$$

$$F(\theta_{t+1}) \leq F(\theta_t) + \langle \nabla F(\theta_t), -\eta \nabla F(\theta_t) \rangle + \eta^2 \frac{L}{2} ||\nabla F(\theta_t)||^2 \tag{87}$$

$$F(\theta_{t+1}) \leq F(\theta_t) - \eta ||\nabla F(\theta_t)||^2 + \eta^2 \frac{L}{2} ||\nabla F(\theta_t)||^2 \tag{88}$$

$$F(\theta_{t+1}) \le F(\theta_t) - \eta ||\nabla F(\theta_t)||^2 (1 - \eta \frac{L}{2}) \tag{89}$$

For stability, $\eta < 2/L$

**ROAST**  Under parameter sharing we consider the following function $\Lambda$ is a diagonal matrix with $\Lambda_{ii} = g(i)\lambda_i$

$$G(\psi) = F(\theta = \Lambda R \psi) \tag{90}$$

$$\begin{aligned} ||\nabla G(\psi_1) - \nabla G(\psi_2)|| &= ||(\Lambda R)^\top (\nabla F(\theta_1 = \Lambda R \psi_1) - \nabla F(\theta_2 = \Lambda R \psi_2))|| \\ &\le \rho(\Lambda R)||(\nabla F(\theta_1 = \Lambda R \psi_1) - \nabla F(\theta_2 = \Lambda R \psi_2))|| \\ &\le \rho(\Lambda R) L_f ||(\Lambda R \psi_1 - \Lambda R \psi_2))|| \\ &\le \rho^2(\Lambda R) L_f ||\psi_1 - \psi_2|| \end{aligned}$$

Thus, $L_g = \rho^2(\Lambda R) L_f$

$$\rho^2(\Lambda R) = \max(\text{eigen-value})(R^\top \Lambda^2 R) = \max_j \sum_{i=1}^{n} \lambda_i^2 \mathbf{1}(h(i) = j) \tag{91}$$

Thus the stability range is $\left(0, \frac{2}{L \max_j \sum_{i=1}^{n} \lambda_i^2 \mathbf{1}(h(i)=j)}\right)$

**STABLE-RPS**  Now let us analyse the gradient scaling algorithm. Again, Consider an update of the gradient descent algorithm, we go from $\psi_t$ to $\psi_{t+1}$ in compressed space which corresponds to $\theta_t = \Lambda R \psi_t$. Where $\Lambda$ and R are as defined above. Also, the gradient scaler is written as a diagonal $m \times m$ matrix $\Gamma$ (this is a different notation than used in main text where it was a vector in $R^m$.)

We have the update equation.

$$\psi_{t+1} = \psi_t - \eta \Gamma \nabla G(\psi_t) \tag{92}$$

Thus,

$$\theta_{t+1} = \theta_t - \eta \left(\Lambda R \Gamma R^\top \Lambda^\top\right) \nabla F(\theta_t) \tag{93}$$

As $F$ is a L-smooth function, we have,

$$F(\theta_{t+1}) \le F(\theta_t) + \langle \nabla F(\theta_t), \theta_{t+1} - \theta_t \rangle + \frac{L}{2} ||\theta_{t+1} - \theta_t||^2 \tag{94}$$

$$F(\theta_{t+1}) \le F(\theta_t) - \eta \nabla F(\theta_t)^\top \left(\Lambda R \Gamma R^\top \Lambda^\top\right) \nabla F(\theta_t) + \frac{\eta^2 L}{2} \nabla F(\theta_t)^\top \left(\Lambda R \Gamma R^\top \Lambda^\top\right)^\top \left(\Lambda R \Gamma R^\top \Lambda^\top\right) \nabla F(\theta_t) \tag{95}$$

$$F(\theta_{t+1}) \le F(\theta_t) - \eta \nabla F(\theta_t)^\top \left(\left(\Lambda R \Gamma R^\top \Lambda^\top\right) - \frac{\eta L}{2}\left(\Lambda R \Gamma R^\top \Lambda^\top\right)^\top \left(\Lambda R \Gamma R^\top \Lambda^\top\right)\right) \nabla F(\theta_t) \tag{96}$$

Consider the matrix,

$$M = \left(\left(\Lambda R \Gamma R^\top \Lambda^\top\right) - \frac{\eta L}{2}\left(\Lambda R \Gamma R^\top \Lambda^\top\right)^\top \left(\Lambda R \Gamma R^\top \Lambda^\top\right)\right) \tag{97}$$

The matrix $\left(\Lambda R \Gamma R^\top \Lambda^\top\right)$ is a $n \times n$ diagonal matrix with $\left(\Lambda R \Gamma R^\top \Lambda^\top\right)_{ii} = \lambda_i^2 \gamma_{h(i)}$ Thus matrix $\left(\Lambda R \Gamma R^\top \Lambda^\top\right)^\top \left(\Lambda R \Gamma R^\top \Lambda^\top\right)$ is also diagonal matrix with $\left(\left(\Lambda R \Gamma R^\top \Lambda^\top\right)^\top \left(\Lambda R \Gamma R^\top \Lambda^\top\right)\right)_{ii} = \lambda_i^4 \gamma_{h(i)}^2$. Thus diagonal element of the matrix in above equation is

$$M_{ii} = \lambda_i^2 \gamma_{h(i)} \left(1 - \frac{\eta L}{2} \lambda_i^2 \gamma_{h(i)}\right) \tag{98}$$

If $\gamma_j = \frac{1}{\sum_{i,h(i)=j} \lambda_i^2}$, and if $\eta < 2/L$, then,

$$\forall i, M_{ii} > 0 \tag{99}$$

as all $\lambda_i > 0$ Since M is a diagonal matrix, M is a positive definite and thus $F(\theta_{t+1}) < F(\theta_t)$ and is stable. Note that the $\gamma$ chosen here is pessimistic (assumes that other $\lambda$'s can be arbitrarily close to 0 and thus performs aggressive scaling. In practice we find that effective update scaler performs well.

## D    PARETO CONTINUITY THEORY

The load factor of each bucket is essentially the number of partitions the original flattened out vector is divided into since each partition adds exactly one element to each bucket. Thus the load $l$ for each bucket is

$$l(i) \leq \left\lceil \frac{n}{m} \right\rceil \tag{100}$$

Which is optimal. When $n = m$, the number of collisions is 0 and thus RPS model and full model are equivalent (except for memory layout of the weights).

The cache fetches can increase if a roast chunk is separated while partitioning. The maximum number of separations is - first it is separated while partitioning and then each of the chunk goes through wrap around (which breaks it into two) - thus $4$. This means maximum additional of 3 cache line fetches that are potentially partially filled.

## E    EXPERIMENTAL DETAILS

**Learning Rate Schedules** We choose the learning rate schedule given by Tanaka et al. (2020) for the milestone steps. We find the the learning rate given in Tanaka et al. (2020) is not the best learning rate even if we use there code base as a starting point of our implementation. So we run the baseline full models with varying learning rate from $\{0.001, 0.01, 0.1, 1, 10\}$ with lr drop-rate of $0.1$ and schedules that are mentioned in table 2. We find that $0.1$ works better for all the full model trainings.

We use the same learning rate schedules and other common hyperparmeters for STABLE-RPS and pruning methods.

Table 2: hyperparameters for both models RESNET20 and VGG11

| Hyperparameter | CIFAR-10 | CIFAR-100 | TINY-IMAGENET |
|:---:|:---:|:---:|:---:|
| Base Learning Rate | 0.1 | 0.1 | 0.1 |
| Milestones | 80,120 | 60,120 | 30,60,80 |
| Learning rate drop | 0.1 | 0.1 | 0.1 |
| batch size | 128 | 128 | 128 |
| total epochs | 160 | 160 | 100 |

**STABLE-RPS hyperparameters** The standard deviation for the RPS array is set to 0.01 for RESNET20 model and 0.05 for VGG11 model. The choice is made following the rule of thumb that scale factors should be in check. However, with gradient scaling, it is not required to be very strict about these parameters

**Pruning iterations** We use 100 iterations for snip and synflow which has been shown to improve its performance. For mag and rand, the number of iterations don't really matter. For GrasP, we tried 100 pruning iterations but that turns out to be bad for the pruning. We thus choose to do only 1 iteration (single shot) for grasp as is originally done.

## F THEOREM VALIDATION.

### F.1 VALIDATION OF THEOREM 1 AND THEOREM 2

The theorem 1 and theorem 2 talk about the theoretical stability region of the SGD algorithm for L-smooth functions under parameter sharing. We validate the theorem with the following experiment.

Consider a parameter sharing setup for the function $f(\theta) = \langle \theta, \theta \rangle$. We know that $L$ for this function is 2. We run simulations with $n = |\theta| = 128$ (recovered full parameter) and $m = |\psi| = 32$ (actual parameter repository). The $\lambda$'s for recovery are drawn from a uniform distribution $[0, \lambda_{max}]$, we vary $\lambda_{max}$ from 0.5 to 2.0. We vary learning rate in increments of 0.1 and see what is the max learning rate where we get convergence.

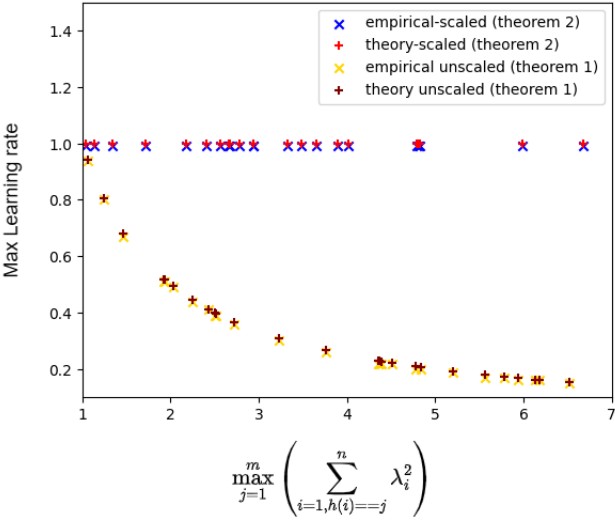

Figure 4: Reduction in stability region due to parameter sharing. (validation of theorem 1 and theorem 2)

The plot shows our empirically computed values of learning rate and theoretically computed values suggested by theorems 1,2. As we can see, the two values are very close to each other validating the stablity region analysis for both unscaled and scaled parameter sharing.

### F.2 VALIDATION OF THEOREM 4

Theorem 4 talks about the inner product preservation under data compression and says that the expected (over data) variance of compression for STABLE-RPS is strictly lower than that for pruning with equality obtained when the data is balanced - all dimensions have distribution with same standard-deviation. In this experiment we consider the following distribution. Given a power-law-coefficient say $c$, the distribution of $X, Y$ is defined as

$$X[i] \sim \mathcal{N}(0, i^{-c})$$

if $c = 0$, then all of them have standard deviation 1 and is the balanced case. The plot shows the expected (over data) variance (over compression) of inner product preservation under the two methods. In this plot the compressed dimension $m = 32$, full dimension is $n = 128$ so it is $4\times$ compression. The plot validates the fact that indeed variance of STABLE-RPS is superior with $c > 0$ and for $c = 0$, it is the same.

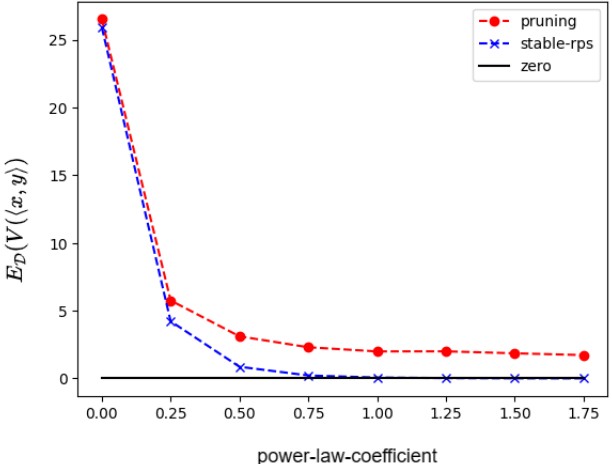

Figure 5: The data $x, y$ is drawn from the following distribution. Given a power-law-coefficient $c$, the distribution of $x, y \sim X$ is defined as $X[i] \sim \mathcal{N}(0, i^{-c})$ (validation of theorem 4)

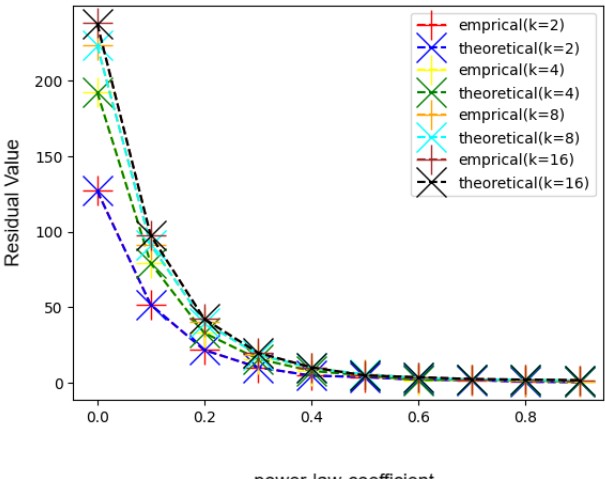

Figure 6: Data is created as follows, $y = \sum_{i=1}^{n} \alpha_i x_i + \epsilon$, $\alpha_i = i^{-c}$ $x_i \sim \mathcal{N}(0, 1)$. $\epsilon \sim \mathcal{N}(0, 0.2)$ c is the power-law coefficient. This plot is created for pruning. $k$ is the compression rate $k = n/m$. (validation of theorem 5)

### F.3 VALIDATION OF THEOREM 5

The theorem 5 represents residual of linear regression in terms of data-compression of correlation vectors. We validate the expression ,

$$Res(\hat{y}) = \sigma_y^2 \left( 1 - \frac{1}{k} \langle \hat{\rho}, \hat{\rho} \rangle \right) \tag{101}$$

The data is created as follows, the data has dimension $n = 256$. The all dimensions are sampled from $\mathcal{N}(0, 1)$. The target is created by following,

$$y = \sum_{i=1}^{n} \alpha_i x_i + \epsilon \tag{102}$$

where $\epsilon \sim \mathcal{N}(0, 0.2)$ and $\alpha_i = i^{-c}$ where $c$ is the power law coefficient. We plot residual under compression for different compression rates $k = \frac{n}{m}$ where $m$ is the size of compressed model. As

shown under different power-law of coefficients (and hence correlations), we see that our theoretical prediction matches empirical observation.

## G EFFECT OF PARETO MAPPING ON LATENCY

The pareto mapping does not follow strict memory alignment of ROAST. However, the theorem 3 states that the effect on cache-miss is not much. We validate this in the following experiment. We consider a model with single linear layer of the size specified in the table. We measure the forward and backward time in ms for this model. This shows that while there is an impact on latency when we go from ROAST to STABLE-RPS, but the impact is minimal. Also, it is much better than HashedNet mapping in which cache-inefficiency is severe in large matrices.

| Model | matrix size : → | 128x128 | 256x256 | 512x512 | 1024x1024 | 2048x2048 | 4096x4096 | 10240x10240 | 20480x20480 |
|---|---|---|---|---|---|---|---|---|---|
| | | FWD + BWD (ms) of a single linear layer under different mappings | | | | | | | |
| **FULL Pytorch** | parameter ↓ | 0.63 | 0.42 | 0.40 | 0.35 | 0.42 | 1.17 | 5.53 | 21.12 |
| **HNet** | 4MB | 0.82 | 0.93 | 0.92 | 0.91 | 1.15 | 2.62 | 10.05 | 38.81 |
| | 32MB | 1.20 | 1.20 | 1.21 | 1.36 | 2.05 | 4.65 | 19.53 | 75.41 |
| | 64MB | 1.56 | 1.55 | 1.58 | 1.81 | 2.83 | 7.31 | 38.23 | 147.58 |
| | 128MB | 2.24 | 2.29 | 2.32 | 2.55 | 3.74 | 9.21 | 47.13 | 183.79 |
| | 256MB | 3.55 | 3.53 | 3.74 | 3.97 | 5.47 | 11.34 | 56.16 | 215.47 |
| | 512MB | 6.63 | 6.66 | 6.80 | 7.12 | 8.66 | 14.82 | 62.13 | 230.48 |
| **ROAST** | 4MB | 1.07 | 0.93 | 0.92 | 0.92 | 1.08 | 1.79 | 6.99 | 26.46 |
| | 32MB | 1.23 | 1.22 | 1.21 | 1.24 | 1.41 | 2.17 | 7.83 | 30.36 |
| | 64MB | 1.59 | 1.57 | 1.57 | 1.74 | 1.78 | 2.59 | 8.66 | 30.01 |
| | 128MB | 2.19 | 2.15 | 2.23 | 2.21 | 2.41 | 3.27 | 9.40 | 31.42 |
| | 256MB | 3.61 | 3.52 | 3.53 | 3.59 | 3.89 | 4.78 | 11.19 | 34.65 |
| | 512MB | 6.68 | 6.65 | 6.66 | 6.71 | 6.95 | 7.90 | 14.31 | 37.77 |
| **STABLE-RPS** | 4MB | 0.92 | 0.93 | 0.92 | 0.92 | 1.12 | 2.27 | 7.86 | 29.92 |
| | 32MB | 1.20 | 1.21 | 1.20 | 1.23 | 1.44 | 2.65 | 8.93 | 33.59 |
| | 64MB | 1.56 | 1.57 | 1.56 | 1.59 | 1.82 | 3.08 | 9.79 | 34.91 |
| | 128MB | 2.16 | 2.24 | 2.20 | 2.20 | 2.57 | 3.71 | 10.40 | 35.58 |
| | 256MB | 3.52 | 3.62 | 3.54 | 3.59 | 3.89 | 5.20 | 12.09 | 37.09 |
| | 512MB | 6.73 | 6.68 | 6.66 | 6.72 | 7.02 | 8.25 | 15.16 | 39.93 |

## H MORE DOMAINS

We ran experiments on PPI dataset (Agrawal et al., 2018) and OGB-ARXIV dataset (Hu et al., 2020) with Graph Attention Network (Velickovic et al., 2017) Model. We see similar observations. We use the implementations provided by the DGL library. Low rank is known to be worse than global mask pruning. We add it here to show comparison.

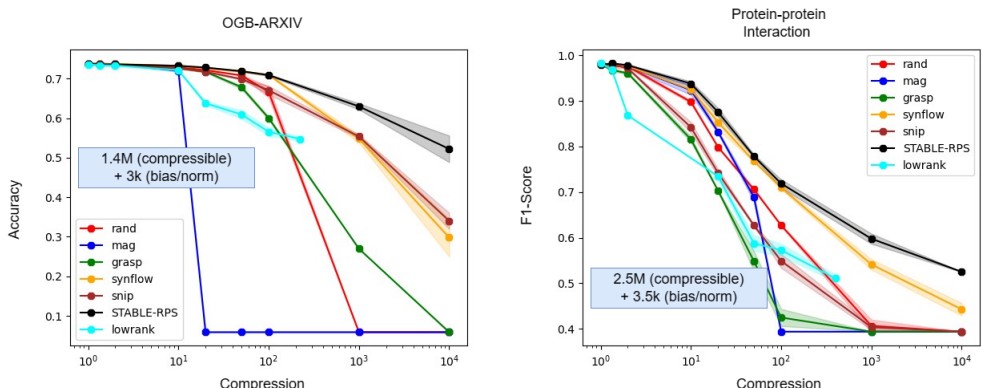

Figure 7: Comparison on graph domain. Low-rank methods cannot obtain more compression than shown in the graph.

Another domain we experiment with is the deep learning recommendation models on click through task. We experiment with Criteo-Kaggle dataset [1] with two models DEEPFM (Guo et al., 2017) and AUTOINT (Song et al., 2019). As most of the parameters in these models come from the embedding tables, we only compress those. The effect of pruning is particularly bad in embedding

---

[1]https://www.kaggle.com/c/criteo-display-ad-challenge

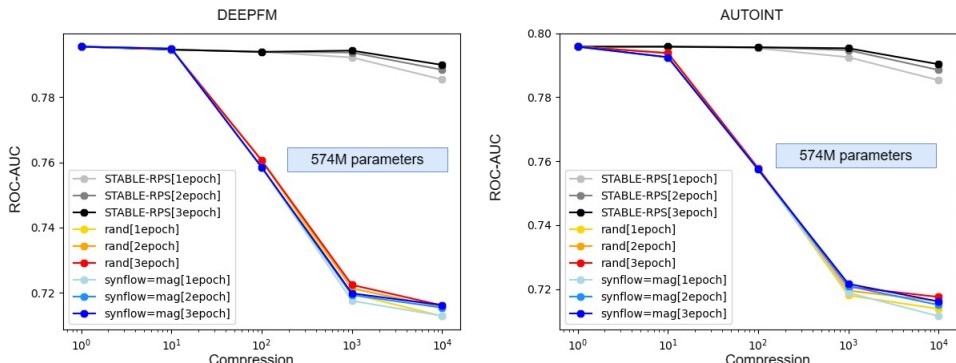

Figure 8: Comparison of pruning vs. RPS on Deep learning recommendation models. Full Model achieves best validation AUC in 1 epoch. The dataset has 40M samples

table compression. We show results of STABLE-RPS, RAND and SYNFLOW which for embedding layer is same as MAG pruning. We also show results at 1, 2 and 3 epochs as they improve with more epochs.

