# OpenReview forum: "In defense of parameter sharing for model-compression"
_ICLR.cc/2024/Conference — ICLR 2024 poster_

### Official Review · Reviewer_sVzo · 2023-10-27

**Soundness:** 2 fair
**Presentation:** 1 poor
**Contribution:** 2 fair
**Rating:** 3
**Confidence:** 4

**Summary:**

This paper looks into the problem of enhancing the quality of compression methods that use parameter sharing. It argues that existing parameter sharing methods, such as ROAST, can have instability issues during initialization and suggests a solution to reduce the instability called STABLE-RPS. The main idea of STABLE-RPS is to reshape the blocks into a single array, split it into several chunks, and apply a hashing function to determine which chunks to share parameters. The paper claims that this approach can improve the stability and pareto-continuity. Experiments on ResNet20/VGG on CIFAR10/100/TinyImagenet demonstrate that the proposed method outperforms several existing pruning methods.

**Strengths:**

While parameter sharing has been heavily studied in the past, the paper made an interesting observation of the shortcomings of existing random parameter-sharing methods in terms of stability and Pareto-continuity. Furthermore, the paper proposes techniques to improve the stability and pareto-continuity of parameter sharing methods.

**Weaknesses:**

1. The writing of the paper lacks clarity and needs some major improvements. For example, the problem statement is vague --- “Is pruning the correct method to reduce model parameters?”. However, the authors do not define what are the “correct methods” for model compression. Are methods that exhibit better stability and Pareto-continuity considered as correct? Are there other correctness conditions?  The paper also uses unconvincing examples to support its arguments. For instance, the authors try to explain pruning adversely affects model capacity, and use an example "Consider pruning a n × d embedding table. If we prune beyond d×, we start getting degenerate zero embeddings." But what does "pruning beyond dx" even mean, and why is not it obvious that pruning reduces model capacity?

2. The paper is narrowly focused on comparing with pruning-based methods, which are only a subset of the model compression techniques. The paper does not justify why pruning methods are the most relevant baselines for the proposed method, which belongs to the parameter sharing paradigm. The paper also ignores other important compression methods, such as quantization, distillation, and low rank factorization.

3. The evaluation setups are weak. Evaluation is done on tiny datasets such as CIFAR10/100 and Tiny-ImageNet and very old architectures such as ResNet-20 and VGG-11 (10-year old), which raise questions on how well the observation from this work can be generalized to larger datasets, such as ImageNet, and more recent architectures, such as vision transformers.

**Questions:**

1. How does STABLE-RPS compare with neural architecture search based methods? Can RPS still outperform search methods in different compression settings?

2. How does the proposed method affect latency? It seems that flattening and chunking blocks would disrupt the original matrix multiplication operations, which could slow down the model execution significantly. The paper should provide latency results for the proposed method and other methods.

3. Why does Theorem 3.2 involve cache line fetches? Cache line fetches are related to execution speed, not memory efficiency. The paper should explain how cache line fetches affect the memory consumption of the proposed method.

---

> ### Author Response · Authors · 2023-11-21
>
> We appreciate your time and efforts to evaluate our paper. We incorporate your suggestions in the updated version and address other concerns below.
>
> **"Is pruning the correct method to reduce model parameters?" What does correct mean?**
>
> We are raising the question that, given an architecture and a parameter memory budget whether "pruning" is the best method to reduce parameters in the model to achieve the memory budget. We have reworded the sentence to read, "Is pruning the optimal method to reduce model parameters?", where optimality is measured in terms of model memory and quality trade-off.
>
>
> **what does pruning beyond dx mean?**
>
> Pruning beyond dx implies we are compressing more than a factor of $d\times$ ( where 2x means reducing the model to half the parameters ).
>
> **why is not it obvious that pruning reduces model capacity**
>
> The model capacity is reduced with any model compression technique. We argue that pruning affects the model capacity more adversely than other methods, such as RPS. The embedding table is a perfect example because compression with pruning leads to degenerate embeddings, which will not happen with RPS, even at very high compression. We have also added datasets in experiments on compressing embedding tables, corroborating this argument.
>
> **why pruning baselines, why not other baselines such as low-rank, distillation, quantization, etc?**
> In this paper, we talk about model compression of a given architecture at the start of training. Pruning is the most advanced and well-studied baseline in this setting.
>
> 1. Low-Rank: Low-rank methods for model compression are inferior to global-mask pruning. We have added a low-rank baseline to our new results on graphs to show this.
>
> 2. Distillation: In this setting, distillation is not the correct baseline as it needs first to train a big model and then use it to train smaller models.
>
> 3. Quantization: Quantization is popularly applied in model compression as either post-quantization or training-aware quantization - both of which are not compression-at-start techniques. To apply quantization at the start, we can train low-precision models, which can be used in conjunction with other techniques, such as pruning and RPS. Also, the compression range of quantization is very limited.
>
> **Diversity of Datasets and models for evaluation**
> 1. We added more datasets from different domains :
> We understand the concern from reviewers about restricted domains of evaluation. To provide more diverse evidence, we have added two more domains of "Graphs with Attention-based Networks"  and "Deep learning Recommendation Models (DLRM) " in Appendix H. The DLRM models (~540M parameters) and the associated dataset are relatively large (40M samples). The observations are consistent in that RPS outperforms other methods. In fact, pruning is especially bad in DLRM models due to the pruning of embedding tables. Pruning of embedding tables at high compression (beyond d\times ) will give degenerate embeddings.
>
>
> 2. Why not train vit on imagenet?
> Creating a single data point by training a vit model on imagenet needs 300 GPU hours if using A100 GPUs. We do not have access to the kind of computing it would need to create Pareto curves for this experiment in a reasonable time.
>
>
> **How does STABLE-RPS compare with neural architecture search-based methods? Can RPS still outperform search methods in different compression settings?**
>
> At this stage, we cannot compare the two methods. NAS finds the best architecture given standard modules. STABLE-RPS and pruning methods find the reduced parameters for a given architecture. From works on pruning, it is clear that once a model is decided, it can be sparsified further and still keep matching accuracies in some cases. However, we still need to choose a good model to begin with, which is where NAS is useful. In the future, NAS can be combined with methods such as STABLE-RPS to obtain even better architectures. But, this direction is out of the scope of the current paper.
>
>
> **How does the proposed method affect latency? Why does Theorem 3.2 involve cache line fetches?**
>
>  In theorem 3, we proved STABLE-RPS introduces minimal cache-inefficiency over ROAST mapping. We add latency numbers to the appendix section G. We measure the training time for different-sized matrices using different mappings in these experiments. We observe that STABLE-RPS mapping introduces a minimal increase in the training time of the model over ROAST.

---

> ### Comment · Reviewer_sVzo · 2023-12-02
> **Response to authors' rebuttal**
>
> Thank you for the reply. The rebuttal does not convince me for the following reasons:
>
> 1. The authors revised the problem formulation as "Is pruning the optimal method to reduce model parameters?"  However, this problem statement seems to be made based on the assumption that pruning is the optimal method for reducing model parameters, which is a false premise. To the reviewer's knowledge, there is few work that claims that pruning is the optimal method for compressing DNN models. In fact, many existing works have demonstrated that there are alternative compression methods that can achieve better capacity-vs-accuracy than pruning, e.g., several existing works have demonstrated that LLMs are better compressed with quantization, as https://arxiv.org/abs/2210.17323.  The reviewer is happy to be convinced if the authors can provide concrete evidence that shows pruning is the optimal method for reducing the model parameters.
>
> 2. The authors claim that "We argue that pruning affects the model capacity more adversely than other methods, such as RPS.". However, this does not seem to make logical sense. Even if existing pruning methods perform empirically worse than RPS, it does not prove convincingly that pruning will in theory lead to worse accuracy than the parameter-sharing method or parameter-sharing method will lead to better pareto optimality, e.g., it seems possible that the model architectures, the datasets, and the training methods used for the tested pruning methods are sub-optimal. The authors do not seem to convincingly exclude the possibility of those factors.
>
> 3. "In this paper, we talk about model compression of a given architecture at the start of training. Pruning is the most advanced and well-studied baseline in this setting." The authors might be referring to zero-shot or single-shot pruning, where pruning is done at the beginning. While this is an interesting line of research, to the reviewer's knowledge, few work has shown that this scheme can lead to better capacity-vs-accuracy results than post-training compression methods or real training speedups. Therefore, the reviewer is not fully convinced that this is a method model scientists can benefit from in practice. Therefore, showing improvements over this baseline limits the generalizability and applicability of the proposed method in practice. Would appreciate it if the authors could provide examples where real-world models are trained when pruning is applied at the beginning of training.

---

### Official Review · Reviewer_L2s3 · 2023-10-31

**Soundness:** 3 good
**Presentation:** 3 good
**Contribution:** 3 good
**Rating:** 5
**Confidence:** 4

**Summary:**

The paper argues towards Randomized Parameter Sharing based models. The authors identified issues and provided solutions in the RPS technique ROAST, regarding stability (ROAST’s sensitivity to initialization hyperparameters, leading to divergence) and Pareto-continuity (ROAST’s inability to recover the accuracy of the original model at zero compression). The authors addressed this by proposing STABLE-RPS. The authors evaluated the method against many SOTA pruning methods and provided a theoretical grounding to their work.

**Strengths:**

Identification and Resolution of Stability, Pareto-Continuity Issues: The authors have identified key issues with existing techniques and proposed STABLE-RPS is an innovative method to address these issues

Rigorous Theoretical Foundation: The authors have also established a strong mathematical foundation to analyze the compression methods. This rigorous approach provides clear insights into how these methods affect vector inner products and under which conditions they perform optimally.

Quality: The work done in the submission is of good quality with clear motivation and clarity. It could be a significant contribution if more empirical evidence is shown by authors.

**Weaknesses:**

Limited Experimental Validation: The paper could benefit more extensive experimental validation to complement the theoretical analysis. The authors only provided experimental evidence on small datasets. Given, that authors claim RPS is the way to go forward it would be good if they can follow up with more experiments

Lack of discussion around additional computation overhead in proposed STABLE-RPS method compared to other methods like ROAST. Some datapoints on what is the end to end training speedups that one can expect with this method will also help.

**Questions:**

Diversity of datasets: Can more experiments be done with different types of datasets and model architectures evaluated to assess the proposed method’s performance?

Some discussion around what are there any specific scenarios where the proposed method might not perform well?

---

> ### Author Response · Authors · 2023-11-21
>
> Thank you for identifying the thoroughness of theory and experiments and the value and soundness of the proposal. Your comment on the work's overall quality, motivation, and clarity is encouraging.
>
> We have incorporated many of your suggestions, as we describe below:
>
> 1. **Additional Experimental validation**
>
>      1. We have added more datasets. : We understand the concern from reviewers about restricted domains of evaluation. To provide more diverse evidence, we have added two more domains of "Graphs with Attention-based Networks" and "Deep learning Recommendation Models (DLRM) " in Appendix H. Specifically, DLRM experiments are relatively large, with the dataset having 40M samples and the model having 540M parameters. The observations are consistent in that RPS outperforms other methods.
>      2. Why not train larger models/datasets such as vit on imagenet? Another reviewer suggested training vit on imagenet. Creating a single data point by training a vit model on imagenet needs 300 GPU hours if using A100 GPUs. We do not have access to the kind of computing it would need to create Pareto curves for this experiment in a reasonable time.
>
> 2. **Computation overhead in the proposed STABLE-RPS method compared to other methods like ROAST**
>
> In theorem 3, we proved STABLE-RPS introduces minimal cache-inefficiency over ROAST mapping. We add latency numbers to the appendix section G.  We measure the training time for different-sized linear layers using different mappings in these experiments. We observe that STABLE-RPS mapping introduces a minimal increase in the training time of the model over ROAST.
>
> 3. **Some discussion around what are there any specific scenarios where the proposed method might not perform well?**
>
> The effectiveness of RPS standalone will depend on how much redundancy exists inside the model given a specific task. For some models, it might be very effective (for example, see the newly added embedding table compression in DLRMs example). In contrast, for some tasks, it will show degradation with compression like the vision tasks we show.
> Compared to methods such as pruning, as pointed out by theory and empirical validation, RPS is superior concerning the memory-accuracy tradeoff.

---

### Official Review · Reviewer_Fae8 · 2023-11-01

**Soundness:** 3 good
**Presentation:** 3 good
**Contribution:** 3 good
**Rating:** 8
**Confidence:** 3

**Summary:**

The authors provide theoretical analysis, algorithmic refinement, and empirical experimentation of randomized parameter sharing (RPS) for model compression, in contrast with the prominent paradigm of parameter pruning for model compression. Guided by their theoretical analysis, the authors’ refined algorithm STABLE-RPS resolves some prior issues with existing RPS methodologies—convergence stability and ability to recover full accuracy at no compression. Further, their large scale empirical analysis reveals that STABLE-RPS can outperform nearly all pruning methods in terms of accuracy-compression tradeoff except for a leading method, lottery ticket rewinding, where STABLE-RPS only outperforms at high compression rates. Research on RPS is motivated by the objective of exploring and improving alternative paradigms for model compression that may be able to deliver better accuracy-compression tradeoff.

**Strengths:**

• Thorough and useful theoretical analysis of existing RPS methods which ultimately inform design of improved RPS alrogithm

• Design and implementation of STABLE-RPS algorithm which resolves two significant issues with existing RPS techniques—convergence stability and Pareto-continuity

• Extensive empirical analysis of STABLE-RPS on three datasets and two architectures compared to seven existing pruning or model compression strategies which demonstrate that STABLE-RPS can outperform nearly all pruning methods in terms of accuracy-compression tradeoff

**Weaknesses:**

• STABLE-RPS cannot outperform a leading pruning method, lottery ticket rewinding, in low to medium compression regime

**Questions:**

1. Is STABLE-RPS compatible with parameter quantization (to achieve additional compression)?

---

> ### Author Response · Authors · 2023-11-21
>
> Thank you for identifying the thoroughness in theory and experiments, and the value and soundness of the proposal.  We would like to let you know that we have added more diverse datasets, latency experiments and validation of theorems to further improve our paper. (details of which are in official comment "Summary of Rebuttal")
>
> We thank you for your encouraging support of the paper.

---

> ### Author Response · Authors · 2023-11-22
> **Is STABLE-RPS compatible with parameter quantization (to achieve additional compression)?**
>
> Yes Quantization can be used in conjunction with other methods including RPS and Pruning to obtain further compression.

---

### Official Review · Reviewer_YEB1 · 2023-11-01

**Soundness:** 3 good
**Presentation:** 2 fair
**Contribution:** 2 fair
**Rating:** 6
**Confidence:** 2

**Summary:**

Stable-RPS is an extension of ROAST, which is a method for sharing parameters using hashing. Parameters are mapped from a shared parameter array into each layer, along with a sign and scaling factor. The paper addresses the stability of ROAST and ensuring that the original performance of the model is maintained at 1x compression (Pareto-continuity). Modifications:

1. A gradient scaling function depending on the scaling factors applied when ROAST maps parameters to layers from the global store
2. A better hash function that ensures Pareto-continuity

The authors then demonstrate that this method of parameter sharing is competitive with contemporary pruning methods.
The authors also include various theoretical results to explain this improved performance.

**Strengths:**

This paper explores missing potential for random parameter sharing in deep neural networks. The early results, such as [Chen et al 2015][hashing], demonstrated competitive performance at the time but the method has received less attention since. It would be valuable for the field to have a comprehensive exploration of the potential of this direction of research.

The improvements to ROAST presented are well motivated and address clear shortcomings of an existing method. They are a good contribution to the field.

The theory investigating random parameter sharing presented in 5 theorems is a useful insight into how random parameter sharing works and will be useful to future work in this area. It seems likely that future research may focus on other alternative hashing methods and will benefit by performing similar analysis.

[hashing]: https://arxiv.org/abs/1504.04788

**Weaknesses:**

Figure 1 fails to explain how ROAST works or how STABLE-RPS relates to ROAST. I don't know what a ROAST array is or what a ROAST++ array is. I don't understand why the resulting Linear array has exactly the same elements (same colors) in both cases.

Equations for ROBE-Z and ROAST are insufficient. I have no way of replicating either method from these descriptions. It looks like integer and modulo division are being used but outside of pseudocode these should be defined with explanation of what they're doing in the equation.

There are 5 theorems stating results, but these results are not checked by experiment. Experimental verification would free the reader from checking the derivation or trusting that it is correct.

Parameter counts for the networks in Figure 3 would be useful, to understand how many parameters the network has at different compression levels.

**Questions:**

Why is there no discussion of the main limitation of random parameter sharing methods versus pruning methods: that they do not reduce the numpy of floating point operations required, while pruning methods do? If pruning methods and RPS perform similarly up to compression factors of 100x then pruning has a significant edge in saving FLOPs. When would RPS really be competitive?

Are there any experimental results on networks other than ResNet-20 or VGG-11? For the same parameter budget there are now many network architectures that perform much better. In other words, where would MobileNet or EfficientNet be placed on Figure 3? Unfortunately, as the compression ratio increases the network quickly enters regions where the accuracy is not worthwhile.

Experimental results are demonstrated on CIFAR-10, CIFAR-100 and Tiny-ImageNet. Would it be possible to explore this method on a contemporary large scale model? What would a scaling law for random parameter sharing look like?

---

> ### Author Response · Authors · 2023-11-21
>
> Thank you for identifying the thoroughness in theory and experiments, and the value and soundness of the proposal. We have incorporated your suggestions into the updated version of the paper.
>
> **Experimental verification for theorems**
>
> Thanks for the great suggestion. We have added empirical verification for theorems in the appendix F. As you can see, the theoretical predictions and empirical computations match in all presented cases.
>
> **Add parameters to the figures**
>
> We have added parameter counts to both existing and newly added figures.
>
> **Would it be possible to explore this method on a contemporary large-scale model?**
>
> 1. We added more datasets from diverse domains: We understand the concern from reviewers about restricted domains of evaluation. To provide more diverse evidence, we have added two more domains of "Graphs with Attention-based Networks"  and "Deep learning Recommendation Models (DLRM) " in Appendix "MORE DOMAINS."  The DLRM models (~540M parameters) and the associated dataset are relatively large (40M samples). The observations are consistent in that RPS outperforms other methods. In fact, pruning is especially bad in DLRM models due to the pruning of embedding tables which is expected since pruning of embedding tables at high compression (beyond d\times ) will give degenerate embeddings.
>
> 2. Why not train larger models/datasets such as vit on imagenet?
> Another reviewer suggested training vit on imagenet. Creating a single data point by training a vit model on imagenet needs 300 GPU hours if using A100 GPUs. We do not have access to the kind of computing it would need to create Pareto curves for this experiment in a reasonable time.
>
>  **Limitation of random parameter sharing methods versus pruning methods, When would RPS really be competitive?**
>
> We mention the current limitations of RPS in the section "Discussions and conclusion,"  Reducing the number of flops in RPS is currently an open problem. We will also mention it in the introduction to avoid any confusion.
>
> There are many situations in which RPS can be potentially beneficial right away.
>
> 1. Embedding heavy models such as DLRMs. We added new results comparing pruning and RPS, and pruning is terrible when compressing embedding tables. As embedding tables do not have any flops, RPS is equally efficient in computation.
>
> 2. MoE models. A mixture of expert models is generally memory-heavy, and their computation is sparsified by design.
>
> 3. Other scenarios exist where higher compression can help us eliminate model parallel execution, which can have latency advantages.
>
> Having said that, this paper focuses on memory-accuracy tradeoff compression techniques. By establishing this, we want to argue for further investigation of RPS to solve existing open problems.
>
> **For the same parameter budget there are now many network architectures that perform much better. In other words, where would MobileNet or EfficientNet be placed on Figure 3?**
>
> Given a memory budget, finding the best architecture is a challenging problem and comes under NAS. At this stage, we cannot compare RPS on one architecture with other architectures (alternatively, we cannot compare the RPS and NAS) . NAS finds the best architecture given standard modules. STABLE-RPS and pruning methods find the reduced parameters for a given architecture. From works on pruning, it is clear that once a model is decided, it can be sparsified further and still keep matching accuracies in some cases. However, we still need to choose a good model to begin with, which is where NAS is useful. In the future, NAS can be combined with methods such as STABLE-RPS to obtain even better architectures. But, this direction is out of the scope of the current paper.
>
>
> **Unfortunately, as the compression ratio increases, the network quickly enters regions where the accuracy is not worthwhile.**
>
>  In practical scenarios, what tradeoff benefits a particular use case depends on the existing resource constraints and quality requirements. Thus, it is impossible to comment on whether RPS will solve the issue to satisfaction. But, in this paper, we want to rigorously compare RPS and pruning paradigms to see which method is better, given the resource constraints, and we find that RPS has better expressivity than pruned models at given memory budget.
>
> **Writing Issues**
> 1. We have rewritten the background on parameter sharing to improve the readability of the section with a focus on the ROAST method. In the appendix, we can add details on other methods, such as ROBE and Hashednet, if needed.
>
> 2. The colors used in the full model denote the different ROAST chunks in the model, and the color in the linear array shows one instance of the mapping -- mapping of one chunk of ROAST or one fold of STABLE-RPS into the array showing which parameter goes where.

---

### Author Response · Authors · 2023-11-21
**Summary of Rebuttal.**

We thank all the reviewers for their time and effort in evaluating our work and suggesting improvements.

**Review and Rebuttal Highlights.**
1. We are excited by the unanimous opinion of all the reviewers regarding the contribution of identifying and resolving existing issues with RPS.

2. We thank reviewers YEB1, Fae8, and L2s3 for recognizing the rigor in theory and its value for the future of the field.

3. We appreciate positive comments from reviewers on the quality of the work from Reviewers L2s3.

4. We thank reviewer Fae8 for their encouraging support to our paper.

**Changes to the draft**

There were suggestions to improve the paper that we have acted on and incorporated into the updated version. Additional sections are added to the appendix.

   1. **Additional experiments (Appendix H)**: Some reviewers asked for diverse empirical evidence of the idea. We have incorporated two domains. The observations on new experiments are consistent with what is presented already in the paper.

    a. Graph Domains, we add results for Graph Attention Network on protein-protein interaction dataset and ogb-arxiv dataset.
    b. Deep learning recommendation models (DLRMs) such as AutoINT and DEEPFM on the Criteo Kaggle dataset; this is a relatively large-scale dataset with 40M samples and models having 540M parameters. This experiment shows a wide gap between RPS and pruning methods, showing the immediate value of RPS.

   2. **Empirical validation of theorems (Appendix F)**: We add empirical validation for our main theorems 1,2,4 and 5 in the paper(Theorem 3 is relatively straightforward) as suggested by YEB1 and L2s3

   3. **Latency evaluation of STABLE-RPS against other hashing methods (Appendix G)**: We add latency results for STABLE-RPS kernels implemented in triton language and compare them against other RPS methods. We see minimal impact on efficiency by using STABLE-RPS over ROAST and STABLE-RPS is much better than HashedNet mapping as expected.

   4. **Writing/presentation improvements (blue text in main paper)**: Acting on the comments by reviewers sVzo and YEB1, we have modified certain word choices and explanations in the paper. We have marked them in blue. Also, we corrected one typo in theorem 2 (removed the sqrt) and simplified theorem 5. Additionally, we have added number of parameters to all of the figures as suggested by YEB1

We will provide detailed individual responses to the reviewers below.

---

### Meta-Review · Area_Chair_zJZF · 2023-12-06

**Metareview:**

The authors study randomized parameter sharing (RPS), a class of techniques used for compressing models. First, they compare a variety of pruning techniques against RPS, concluding that there are regimes where RPS beats out pruning. Next, they examine the best-in-class RPS technique and observe certain weaknesses, which they address with a variant. They analyze this variant theoretically and through a series of experiments.

Overall this is strong work. There's a general need for this type of analytical work that has both extensive empirical comparisons but also analysis of the strengths and weaknesses of state-of-the-art techniques.

The reviewers largely agreed. One reviewer asked for a number of clarifications, which the authors mostly addressed---further strengthening the paper.

Overall, while the scores for this paper are borderline, I believe there is a nice contribution here that advances our state of knowledge for an important class of techniques.

**Justification For Why Not Higher Score:**

The contribution here is to a particular technique that, while important, is probably not going to have the impact that I would expect for papers that are going to be recognized with spotlights or orals.

**Justification For Why Not Lower Score:**

Generally this type of work (examining what works well and what doesn't for the best-in-class technique in a practical area, using both a theoretical and practical lens) is worth accepting.

---

### Decision · Program_Chairs · 2024-01-16

Accept (poster)